# DR. BENCH: A Multidimensional Evaluation for Deep Research Agents, *from Answers to Reports*

## Abstract

As an embodiment of intelligence evolution toward interconnected architectures, Deep Research Agents (DRAs) systematically exhibit the capabilities in task decomposition, cross-source retrieval, multi-stage reasoning, information integration, and structured output, which markedly enhance performance on complex and open-ended tasks. However, existing benchmarks remain deficient in evaluation dimensions, response format, and scoring mechanisms, limiting their effectiveness in assessing such agents. This paper introduces **DR. BENCH**, a multidimensional evaluation framework tailored to DRAs and long-form report-style responses. The benchmark comprises 214 expert-curated challenging tasks across 10 broad domains, each accompanied by manually constructed reference bundles to support composite evaluation. This framework incorporates metrics for semantic quality, topical focus, and retrieval trustworthiness, enabling a comprehensive evaluation of long reports generated by DRAs. Extensive experimentation confirms the superior performance of mainstream DRAs over web-search-tool-augmented reasoning models, yet reveals considerable scope for further improvement. This study provides a robust foundation for capability assessment, architectural refinement, and paradigm advancement of DRAs.

## 1. Introduction

The core architecture of AI paradigms is undergoing a transition from closed, static, parameter-driven Large Language Models (LLMs) to active, cognitive, and interconnected agent systems endowed with external perception and integrative mechanisms (Wang et al., 2024; Zhang et al., 2025a).

[1]Anonymous Institution, Anonymous City, Anonymous Region, Anonymous Country. Correspondence to: Anonymous Author <anon.email@domain.com>.

Preliminary work. Under review by the International Conference on Machine Learning (ICML). Do not distribute.

Amid growing demands for heterogeneous information acquisition and increasing task complexity, Deep Research has gradually emerged as a paradigm of agent systems characterized by cognitive planning and integrative capabilities (Xu & Peng, 2025), which enable autonomous task decomposition, cross-source retrieval, multi-step reasoning, and structured expression, thereby substantially enhancing model adaptability and expressive performance in real-world applications (Huang et al., 2025; Zhang et al., 2025b).

Although Deep Research Agents (DRAs) demonstrate strong task execution capabilities, the current benchmarking framework remains significantly outdated in both design philosophy and coverage scope. First, existing benchmarks predominantly target discrete short-text outputs, such as multiple-choice answers or brief phrases (Wei et al., 2024; Zhou et al., 2025; Ho et al., 2020; Yang et al., 2018), which, although conducive to efficient evaluation and automation, cannot be extended to complex report-style generation tasks and fail to reflect the demands of DRAs for logical inference and linguistic organization. Second, most benchmarks continue to assess isolated competencies, focusing primarily on reasoning or web search, without establishing systematic criteria for evaluating integrated performance. In particular, mechanisms for assessing citation authority, source validity, and semantic drift in long-form outputs remain absent. Mainstream evaluation methods rely either on string matching (Cohen et al., 2025; Monteiro et al., 2024), which fails to capture semantic adequacy, or on similarity scoring with LLMs as judgers (Chen et al., 2025a; Pham et al., 2025), which lacks transparent and verifiable standards and is therefore prone to subjectivity and instability.

To address the limitations inherent in existing evaluations, our study proposes a rigorous benchmark, authored by human experts and specifically designed for DRAs, which targets high-difficulty and high-precision tasks with the goal of systematically assessing the overall performance in report-style long-text generation. On this basis, we develop a multidimensional evaluation framework intending to accurately and stably measure the quality and credibility of generated reports. Our research yields three key contributions:

(1) We introduce the **D**(eep)**R**(esearch). **BENCH** paired with manually constructed reference bundles compris-

ing query-specific and general-report rubrics, trustworthy sources, focus-anchor and focus-deviation keywords. Covering 214 challenging report-oriented entries across 10 broad domains, DR. BENCH rigorously and efficiently enables broad and granular characterization of DRA tasks, which addresses existing deficiencies in evaluation dimensions and response formats.

(2) We develop a systematic and multidimensional evaluation framework for long-form report-style outputs, which captures key processes of DRAs such as task reasoning, information retrieval, content synthesis, and structured articulation. By jointly modeling semantic quality, topical focus, and retrieval trustworthiness, our framework overcomes limitations of conventional methods and demonstrates high transferability to long-text generations beyond DRAs.

(3) We conduct large-scale experiments involving five mainstream DRAs, one advanced agent model, and seven reasoning models enhanced with web-search tools. Quantitative results indicate that DRAs consistently outperform tool-augmented models in overall task execution proficiency and report generation quality, while revealing persistent limitations in architectural paradigms and behavioral mechanisms that warrant further refinement and continued development.

## 2. Related Works

### 2.1. Deep Research Agents

Early LLMs, such as GPT-3 (Brown et al., 2020) and PaLM (Chowdhery et al., 2023), rely on static training corpora and closed parameter spaces, while their knowledge is entirely derived from the training phase and cannot be updated or supplemented during inference. To overcome the epistemic constraints of static language models, researchers have explored mechanisms for integrating LLMs with external tools, giving rise to the paradigm of Tool-Augmented LLMs, exemplified by GPT-4 (Achiam et al., 2023), Gemini 1.5 (Google, 2024), Claude 3 (Anthropic, 2024), Toolformer (Schick et al., 2023), and Qwen 1.5 Agent (Alibaba, 2024). These models leverage external interfaces such as web browsers to enable dynamic information acquisition and cross-modal perception, reflecting a shift from knowledge encapsulation toward tool-mediated cognition.

Beyond generative and inferential capabilities, DRAs integrate cognitive planning and information fusion to support end-to-end workflows for complex and open-ended tasks. Their functionality encompasses task recognition, multi-stage decomposition, heterogeneous retrieval, cross-source aggregation, and structured report generation, emphasizing system-level autonomy and procedural integrity.

This paradigm is broadly applicable to academic research, policy analysis, technology evaluation, and market intelligence, while remaining applicable to daily-life scenarios for general users. Representative systems include open-source DRAs like Tongyi DeepResearch (Alibaba, 2025) and commercial platforms such as Grok Deep Search (xAI, 2025), Sonar Deep Research (Perplexity, 2025), and o3 Deep Research (OpenAI, 2025), which implement full-spectrum research pipelines. These systems exemplify a shift from static knowledge encapsulation to cognitively extended intelligence, advancing LLMs toward agentic architectures with strategic reasoning capabilities.

### 2.2. Existing Benchmarks

With the growing adoption of Tool-Augmented LLMs, both academia and industry have increasingly focused on their performance in web-search tasks. Existing benchmarks, including GAIA (Mialon et al., 2023), WebWalker (Wu et al., 2025), BrowseComp (Wei et al., 2025), WideSearch (Wong et al., 2025), BrowseComp-Plus (Chen et al., 2025b), and Deep Research Bench (Bosse et al., 2025), primarily evaluate LLMs using closed-form queries that require verifiable short answers to facilitate automated scoring and alignment. However, they lack coverage of key behaviors such as task decomposition, cross-source retrieval, and structured synthesis, and provide limited assessment of content hierarchy, discourse structure, and information integration. A rough comparison of existing benchmarks can be found in Appendix F. In addition, most rely on surface-level matching or similarity metrics, such as Exact Match, BLEU (Papineni et al., 2002), ROUGE (Lin, 2004), and BERTScore (Zhang et al., 2019), which struggle to capture semantic depth and structural fidelity, thereby limiting their effectiveness in evaluating the true capabilities of DRAs.

Contemporaneous work has begun to explore evaluation methods tailored to report-style outputs. DeepResearch Bench (Du et al., 2025) is the first to assess reference answers alignment and retrieval. However, it depends heavily on static reference reports, making it difficult to accommodate evolving query expectations. Its automated rubrics lack contextual sensitivity and focus on generic surface, failing to reflect human preferences for report quality and structure. Its retrieval evaluation emphasizes consistency between statements and cited links but overlooks the credibility of sources. Benchmarks such as ResearchQA (Yifei et al., 2025), DeepResearch Arena (Wan et al., 2025), and ReportBench (Li et al., 2025) constructed 21K, 10K, and 0.6K academic tasks respectively. These benchmarks also rely on automatically generated rubrics that exhibit limited stability and interpretability, which undermines their reliability for high-precision judgment. Additionally, large-scale benchmarks increase evaluation costs, especially for expensive DRAs, reducing their practical utility.

Overall, existing benchmarks fail to rigorously and comprehensively evaluate report-style long-form outputs in alignment with human expectations. They lack precise rubrics and trustworthy references, making it difficult to systematically characterize the full capabilities of DRAs.

## 3. DR. BENCH

### 3.1. Domains

The DR. BENCH was meticulously constructed and repeatedly validated by human experts, comprising 214 high-complexity entries across diverse domains. It is designed to challenge existing DRAs in task understanding, decomposition, execution, and aggregation. Based on semantic relevance, entries are systematically categorized into ten broad domains, with detailed distribution provided in Figure 1 and Appendix A. The dataset exhibits a high degree of professionalism and diversity in its design, effectively simulating complex and dynamic queries encountered in real-world scenarios, and demonstrating broad coverage and strong representativeness.

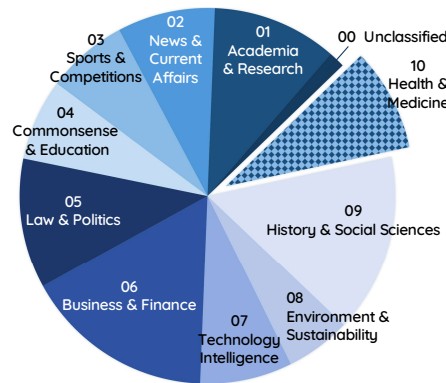

*Figure 1.* The proportion of DR. BENCH entries across domains.

### 3.2. Components

Each entry consists of a query instruction paired with a reference bundle designed to support performance evaluation. The bundle comprises five core modules: Query-Specific Rubrics (QSRs), General-Report Rubrics (GRRs), Trustworthy-Source Links (TSLs), Focus-Anchor Keywords (FAKs), and Focus-Deviation Keywords (FDKs), each of which corresponds to a distinct capability that a DRA is expected to demonstrate.

#### 3.2.1. QUERY

As the core input to DRAs, queries are designed to elicit structured long-form reports rather than traditional short, discrete answers, while embodying diversity and representativeness by (1) covering topics from ancient history to contemporary events, including both long-term technological

evolution and short-term dynamic shifts; (2) encompassing major countries and regions, with systematically designed cross-regional comparative tasks; (3) involving diverse disciplines, with many tasks intentionally designed for interdisciplinary integration; (4) entailing structured texts across various styles including academic writing, data-driven analysis, technical deconstruction, strategic planning, policy evaluation, and event reconstruction; (5) incorporating multi-level causal chains and cross-source information fusion; and (6) including both quantitative and qualitative analysis.

To enhance consistency and reproducibility, DR. BENCH systematically incorporates spatiotemporal robustness into the design of queries and rubrics. For fact-based tasks, each query explicitly defines temporal and geographic boundaries to mitigate external fluctuations in response content. For open-ended tasks, rubrics emphasize logical structure and reasoning validity to ensure generalizability and adaptability. All queries adopt the instruction "write a report/essay/..." to unify output format and reinforce task orientation. Overall, the query design achieves a high degree of semantic complexity, task diversity, and evaluative operability.

#### 3.2.2. QUERY-SPECIFIC RUBRICS

QSRs are designed to evaluate the task completion quality. Each QSR is custom-built by experts based on corresponding query, reflecting human expectations and evaluative preferences regarding factual accuracy and logical validity. Scoring follows a clearly defined binary (Yes/No) or ternary (Partial) scheme to ensure consistency and operational clarity. Each query is equipped with at least 8 QSRs, with a total score of 30, assigned in accordance with task importance.

QSRs are deeply embedded within the semantic structure of each task, offering high alignment and diagnostic precision. Their design spans core dimensions such as information coverage, mechanism explanation, structured expression, semantic precision, source verification, evidence organization, heterogeneity analysis, methodological transparency, temporal logic, and interdisciplinary integration. Appendix G outlines the core dimensions of QSRs design together with their corresponding descriptions. The QSRs provide both theoretical grounding and practical guidance for evaluating DRAs performance, while also laying a structural foundation for future automated scoring systems.

#### 3.2.3. GENERAL-REPORT RUBRICS

GRRs assess the quality of structured expression. Independent of specific queries, GRRs evaluate reports from a general perspective using binary judgment across 7 key dimensions, namely structural organization, logical clarity & expression, informational coverage & content depth, citation quality & source credibility, originality & insight, data usage & analytical rigor, and formatting consistency.

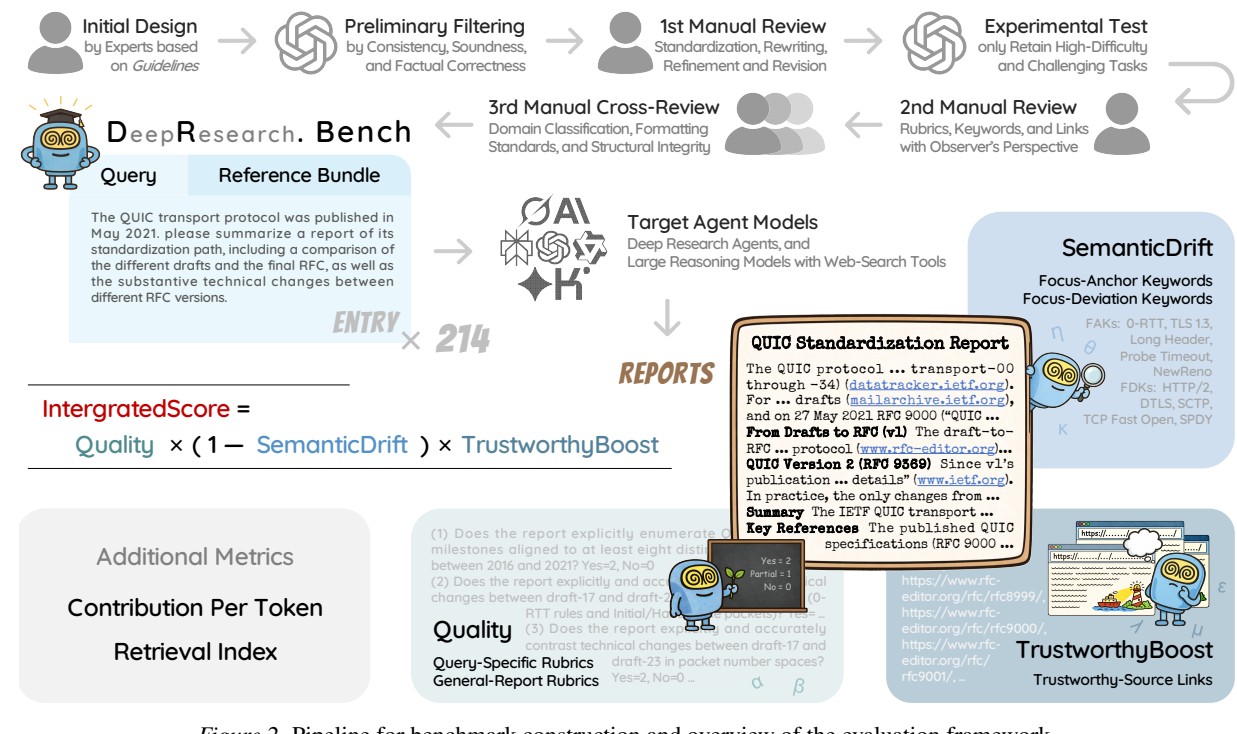

*Figure 2.* Pipeline for benchmark construction and overview of the evaluation framework.

GRRs constitute a necessary, objective, and universally applicable set of standards that delineate the fundamental human expectations for report-style responses: a qualified report must not only accomplish the assigned task but also satisfy general quality requirements such as structural completeness, logical coherence, and insightful argumentation. Comprising 48 rubrics with a total score of 73, the design emphasizes generalizability, normative clarity, and expressive strength, thereby providing a unified quality benchmark for performance evaluation across tasks and models, and establishing a comprehensive evaluation framework grounded in the dual pillars of content and expression, as outlined in Appendix B.

### 3.2.4. TRUSTWORTHY-SOURCE LINKS

TSLs serve as indicators of the trustworthiness of heterogeneous retrieval and cross-source aggregation. They are designated by experts in variable quantities, and consist of durable, stable, and reliable website links that are authoritative, official, accessible, and contain original information necessary to answer the query. Subjective or non-primary sources such as forums and blogs are excluded. Each link is precisely anchored to the specific page containing the target information, ensuring accuracy, verifiability, and confidence in information acquisition.

To prevent TSLs expiration, we (1) defined stability and longevity requirements for queries and TSLs during data construction, mandating that TSLs originate from author-

itative, official, and primary sources (e.g., government or academic institutions) that are reliably accessible and resistant to change, thereby ensuring stability from the outset; (2) introduced hostname matching at the metric mechanism to reduce dependence on specific pages, and incorporated hyperparameters in TrustworthyBoost, enabling full-match stability to be adjusted across evaluation targets; and (3) established long-term maintenance through regular link validity checks (e.g., quarterly) and the planned use of snapshots as permanent references.

### 3.2.5. FOCUS-ANCHOR KEYWORDS

FAKs are used to evaluate thematic focus during cross-source content aggregation. Each set of FAKs is specified by expert designers for the given query and consists of 5 semantically stable core terms, while avoiding superficial phrasing from the query itself. FAKs serve to assess the thematic focus and key point coverage of generated content, enabling effective evaluation in terms of semantic consistency and analytical depth.

### 3.2.6. FOCUS-DEVIATION KEYWORDS

FDKs are used to evaluate the degree of thematic drift. Each set consists of five terms that are prone to triggering topic divergence. Their presence typically indicates that the generated content has deviated from the original query focus, resulting in reduced semantic coherence and increased informational noise. In high-cost DRA processes, such devi-

ations lead to unnecessary consumption of resources, ultimately compromising overall performance.

For example, when a query focuses on "men's football matches", the term "women" would divert the report off-topic. Similarly, when the query concerns the "electric vehicles", references to "fuel cars" or "motorcycles" are classified as FDK. During data construction, these terms were annotated by experts and validated through multiple reviews to ensure their consistency and appropriateness.

### 3.3. Construction Pipeline

To ensure the difficulty, quality, and semantic validity, DR. BENCH adopts a multi-stage review pipeline for systematic verification and refinement of expert-generated entries. This process integrates manual design, machine auditing, and cross-validation to establish a highly granular framework for data generation and quality control, as illustrated in the upper half of Figure 2.

In the initial stage, experts design diverse data units based on the unified *Construction Guide* and their own expertise within predefined domains. The units then undergo LLMs auditing to detect semantic inconsistencies, logical errors, and factual inaccuracies. During the 1st round of manual review, QSRs are verified for validity, with rubric criteria refined. For binary items with intermediate cases, a clearly defined "Partial" option is introduced to enhance scoring granularity and consistency. After preliminary filtering, queries and QSRs are standardized and rewritten to ensure stylistic and structural correctness. TSLs and Keywords are also supplemented and revised at this stage. The revised entries then undergo LLM-based difficulty testing, followed by the 2nd round of manual review, which focuses on evaluating QSRs from observers' perspective and assessing the anchoring effectiveness of keywords. Each link is individually verified for accessibility and authority. The 3rd round of cross-review conducts a comprehensive quality check, covering domain classification, formatting standards, and structural integrity. Examples can be found in Appendix C.

This multi-phase mechanism significantly improves accuracy, consistency, and reproducibility, while also reducing subjective bias and annotation errors, providing a robust foundation for model evaluation.

## 4. Evaluation Framework

Built on semantic quality, topical focus, and retrieval trustworthiness, a structured and multidimensional evaluation framework is proposed for report-style generation tasks, featuring an integrated scoring system inspired by principles from statistics and operations. It prioritizes transparency, interpretability, and practical utility for robust evaluation of complex tasks, as illustrated in the lower half of Figure 2.

### 4.1. Semantic Quality

Semantic quality evaluates the overall performance of response reports in terms of task completion and general quality. It integrates scores from both QSRs and GRRs. Drawing on the Weighted Average Method (WAM) from Multi-Attribute Decision Making (MADM), the metric applies ratio normalization to both scores and assigns weighting coefficients $\alpha$ and $\beta$, where $\alpha + \beta = 1$, to construct a fused model of multidimensional quality signals. The calculation formula of Quality $\in [0, 1]$ is as follows:

$$\text{Quality} = \alpha \cdot \mathcal{N}_{\text{Ratio}} \left[ \sum_{i=1}^{N} \text{QSR}_{\text{score}}^{(i)} \right] + \beta \cdot \mathcal{N}_{\text{Ratio}} \left[ \sum_{j=1}^{M} \text{GRR}_{\text{score}}^{(j)} \right]$$

where $\text{QSR}_{\text{score}}^{(i)}$ denotes the $i$-th QSR score; $\text{GRR}_{\text{score}}^{(j)}$ refers to the $j$-th GRR score; $N$ and $M$ represent the number of QSRs and GRRs respectively; $\mathcal{N}_{\text{Ratio}}[\cdot] = \cdot / \sum_i \text{FullScore}(r_i)$ denotes the ratio normalization; $\alpha$ and $\beta$ are weighting parameters reflecting the relative importance in the overall quality assessment.

### 4.2. Topical Focus

Topical focus is evaluated through the SemanticDrift metric, which jointly considers the absence of FAKs and the misuse of FDKs to measure the degree of thematic deviation.

$\text{FAK}_{\text{Drift}} \in [0, 1]$ quantifies the omission of core keywords. The frequency of each keyword in the report is scaled by its expected value $\epsilon$, and a minimum function is introduced to implement a threshold control. A FAK is penalized when its frequency falls below $\epsilon$, thereby enforcing the requirement for substantive keyword presence and ensuring that the score reflects semantic completeness rather than accidental occurrence. Drawing on the TF × IDF paradigm from information retrieval, $\text{FAK}_{\text{Drift}}$ is constructed as the product of frequency component and semantic relevance:

$$\text{FAK}_{\text{Drift}} = 1 - \frac{1}{K} \sum_{k=1}^{K} \left[ \min \left( \frac{\text{freq}^{(k)}}{\epsilon_+}, 1 \right) \times \mathcal{N}_{\text{Ratio}}(\text{rele}^{(k)}) \right]$$

where $\text{freq}^{(k)}$ is frequency of the $k$-th FAK; $\text{rele}^{(k)} \in \{1, 2, 3, 4, 5\}$ is the semantic relevance assessed by LLMs; $K$ represents the number of FAKs; $\epsilon_+$ is the expectation scaling factor. A higher $\text{FAK}_{\text{Drift}}$ value indicates weaker thematic focus and insufficient coverage of core concepts in the report. Similarly, $\text{FDK}_{\text{Drift}} \in [0, 1]$, which quantifies the intensity of thematic distraction, is defined as follows:

$$\text{FDK}_{\text{Drift}} = \frac{1}{L} \sum_{l=1}^{L} \left[ \min \left( \frac{\text{freq}^{(l)}}{\epsilon_-}, 1 \right) \times \mathcal{N}_{\text{Ratio}}(\text{rele}^{(l)}) \right]$$

where $\text{freq}^{(l)}$ is frequency of the $l$-th FDK; $\text{rele}^{(l)} \in \{1, 2, 3, 4, 5\}$ is the relevance score; $L$ represents the number of FDKs; $\epsilon_-$ is the scaling factor. A higher $\text{FDK}_{\text{Drift}}$ indicates stronger thematic deviation. The SemanticDrift is a weighted combination of $\text{FAK}_{\text{Drift}}$ and $\text{FDK}_{\text{Drift}}$:

$$\text{SemanticDrift} = \lambda \cdot \text{FAK}_{\text{Drift}} + \mu \cdot \text{FDK}_{\text{Drift}} \in [0, 1]$$

where $\lambda + \mu = 1$, reflecting the relative importance. SemanticDrift reflects the degree of thematic deviation in the generated report, with higher values indicating weaker alignment to the intended topic and lower values suggesting stronger semantic focus and consistency.

### 4.3. Retrieval Trustworthiness

Retrieval Trustworthiness evaluates the credibility of external information retrieval and usage in response reports. Modeling based on the hit rate of TSLs and using confidence enhancement mechanism inspired by multiplicative fusion in Bayesian updating, this approach transforms match rates into multiplicative scoring factors to increase the evaluative weight of citation quality. Specifically, matches are categorized into full matches and hostname matches, where $\text{Rate}_{\text{full hit}}$ serves as a recall-like metric capturing the precise coverage of provided TSLs, and $\text{Rate}_{\text{host hit}}$ reflects the proportion of generalized mentions whose annotation links share the same source domains as the recommended references. These two are assigned different weights and combined to compute the TrustworthyBoost $\in [1, 1 + \eta]$ as:

$$\text{TBoost} = 1 + \eta \cdot \left[ \theta \cdot \underbrace{\left( \frac{\text{match}_{\text{full}}}{S} \right)}_{\text{Rate}_{\text{full hit}}} + \kappa \cdot \underbrace{\left( \frac{\text{match}_{\text{host}}}{T + 1} \right)}_{\text{Rate}_{\text{host hit}}} \right]$$

where $\text{match}_{\text{full}}$ indicates the number that have been exactly matched in TSLs; $\text{match}_{\text{host}}$ refers to the number of annotations sharing the same hostname as the TSLs; $S$ and $T$ denote the sizes of the TSLs and the annotations respectively; $\theta$ and $\kappa$ represent the weights for full and hostname matches respectively, with $\theta + \kappa = 1$. The coefficient $\eta$ controls the magnitude of the confidence boost, thereby preventing excessive inflation due to high confidence and avoiding complete nullification when confidence is low. This mechanism preserves scoring stability while enhancing evaluative sensitivity to verifiability and source reliability, thereby improving the responsiveness to external evidence signals.

### 4.4. Integrated Scoring Framework

The integrated metric $\in [0, 120]$ adopts a multiplicative weighting model to enable multidimensional evaluation of report-style generation tasks.

$$\text{IntegratedScore} = \text{Quality} \times (1 - \text{SDrift}) \times \text{TBoost}$$

Here, Quality assesses the structural integrity and content quality of the report, SemanticDrift reflects the degree of thematic deviation and is transformed into a positive factor via $1 - \text{SemanticDrift}$, and TrustworthyBoost enhances the credibility weight based on authoritative link coverage. This design logic penalizes semantic drift and rewards external support, producing a normalized score on a 100-point scale.

---

**Algorithm 1** Multidimensional Evaluation Framework

---

1: **for** `entry` in `entries` **do**
2:     Extract `query`, `QSRs`, `GRRs`, `TSLs`, `FAKs`, `FDKs`
3:     Extract `report`, `annotations`, `token_usage`
4:     ***# Semantic Quality***
5:     $\text{QSR}_{\text{scores}} \leftarrow \sum \text{QSR}_{\text{score}}$ from LLM-Judger over `QSRs`
6:     $\text{GRR}_{\text{scores}} \leftarrow \sum \text{GRR}_{\text{score}}$ from LLM-Judger over `GRRs`
7:     $\text{Quality} \leftarrow \alpha \cdot \mathcal{N}_{\text{Ratio}}(\text{QSR}_{\text{scores}}) + \beta \cdot \mathcal{N}_{\text{Ratio}}(\text{GRR}_{\text{scores}})$
8:     ***# Retrieval Trustworthiness***
9:     $\text{Rate}_{\text{full hit}} \leftarrow \text{Match}_{\text{full}}/|\text{TSLs}|$
10:     $\text{Rate}_{\text{host hit}} \leftarrow (\text{Match}_{\text{host}} - \text{Match}_{\text{full}})/|\text{annotations}+1|$
11:     $\text{TrustworthyBoost} \leftarrow 1 + \eta \cdot (\theta \cdot \text{Rate}_{\text{full hit}} + \kappa \cdot \text{Rate}_{\text{host hit}})$
12:     ***# Topical Focus***
13:     **for** `FAK` in `FAKs` **do**
14:         relevance $\leftarrow$ LLM-Judger(`FAK`, `report`)
15:         frequency $\leftarrow$ count of `FAK` in `report`
16:         $\text{FAK}_{\text{score}} \leftarrow \min(\text{frequency}/\epsilon_+, 1) \cdot \mathcal{N}_{\text{Ratio}}(\text{relevance})$
17:     **end for**
18:     $\text{FAK}_{\text{Drift}} \leftarrow 1 - \sum \text{FAK}_{\text{score}}/|\text{FAKs}|$
19:     Perform same procedure for `FDKs` to compute $\text{FDK}_{\text{Drift}}$
20:     $\text{SemanticDrift} \leftarrow \lambda \cdot \text{FAK}_{\text{Drift}} + \mu \cdot \text{FDK}_{\text{Drift}}$
21:     ***# Integrated Scoring Framework***
22:     $\text{InteScore} \leftarrow \text{Quality} \cdot (1 - \text{SDrift}) \cdot \text{TBoost} \cdot 100$
23:     $\text{ContriPerToken} \leftarrow \text{InteScore}/(\text{token}_{\text{total}} - \text{token}_{\text{input}})$
24: **end for**
25: ***# Aggregate Metrics***
26: $\overline{\text{InteScore}} \leftarrow$ average of InteScore over entries
27: $\overline{\text{ContriPerToken}} \leftarrow$ average of ContriPerToken over entries

---

As shown in Algorithm 1, the framework addresses limitations in traditional evaluation methods when applied to DRAs. It offers strong scalability and transferability, making it broadly applicable to performance assessment of structured long-text generation tasks, particularly those involving tool-augmented systems.

### 4.5. Additional Metrics

To enable a more comprehensive assessment of DRAs ability, we design a set of supplementary metrics based on accessible response metadata to characterize model performance and efficiency in real-world execution, including token consumption, number of reasoning steps, and the volume of links involved during retrieval.

$$\text{ContributionPerToken} = \frac{\text{IntegratedScore}}{token_{\text{total}} - token_{\text{input}}}$$

To evaluate cost-effectiveness under limited resources, the Contribution/Token metric evaluates model efficiency based on actual token expenditure in reasoning and generation,

Table 1. Results of the main experiment, with the **highest score** in bold and the second-highest underlined in each column.

| Models | Quality ↑ | 1−SDrift ↑ | TBoost ↑ | InteScore ↑ | Usage | C/Token ↓ |
|---|---|---|---|---|---|---|
| Qwen-deep-research | 0.6348 | 0.5248 | 1.0288 | **34.6480** | 9258 | 0.0100 |
| Sonar-deep-research | 0.6184 | **0.5271** | 1.0238 | 33.4668 | 8254 | 0.0043 |
| o3-deep-research-2025-06-26 | 0.6176 | 0.5184 | 1.0171 | 32.9004 | **25038** | 0.0014 |
| Kimi-K2-0905-preview | **0.6707** | 0.4671 | 1.0153 | 32.0651 | 2079 | 0.0164 |
| Grok-4-0709-search | 0.6130 | 0.4890 | 1.0283 | 31.3490 | 3012 | 0.0112 |
| Gemini-2.5-pro | 0.5506 | 0.4856 | 1.0130 | 27.3364 | 5446 | 0.0072 |
| o4-mini-deep-research-2025-06-26 | 0.5666 | 0.4803 | 1.0203 | 28.0391 | 18640 | 0.0016 |
| GPT-5-2025-08-07 | 0.5560 | 0.4593 | **1.0383** | 27.3312 | 7006 | 0.0045 |
| GPT-4o-search-preview-2025-03-11 | 0.4945 | 0.4496 | 1.0073 | 22.5645 | 1005 | **0.0247** |
| GPT-4.1-2025-04-14 | 0.4762 | 0.4694 | 1.0027 | 22.4382 | 1252 | 0.0194 |
| Claude-opus-4-1-20250805 | 0.4559 | 0.4674 | 1.0202 | 22.0047 | 2267 | 0.0101 |
| Claude-sonnet-4-20250514 | 0.4491 | 0.4735 | 1.0184 | 21.7235 | 2267 | 0.0097 |
| Claude-3-7-sonnet-20250219 | 0.3996 | 0.4737 | 1.0148 | 19.3415 | 2327 | 0.0084 |

effectively measuring the information density per token. When long texts maintain high information density, efficiency scores can be high; they are truly lowered only when the text contains substantial redundancy or off-topic content, which is precisely what we aim to detect. This metric is designed to encourage models to produce efficient outputs.

$$\text{RetrievalIndex} = \frac{num_{\text{annotated}}}{num_{\text{retrieved}} + 1} \in [0, 1]$$

RetrievalIndex evaluates the filtering and aggregation capability of DRAs during the retrieval process. It is defined as the ratio between the number of annotations ultimately adopted in the report and the total number of links retrieved during the search phase, reflecting the model's ability to effectively distill valuable content from large-scale information. A lower index indicates stronger selectivity and aggregation, suggesting that the DRA can extract more relevant and informative content from redundant sources, thereby enhancing the specificity and density of the generated output.

## 5. Experiments

### 5.1. Overview

We evaluated a total of thirteen models under DR. BENCH, including five DRAs (`o3-deep-research-2025-06-26`, `qwen-deep-research`, `sonar-deep-research`, `grok-4-0709-search`, and `o4-mini-deep-research-2025-06-26`), one advanced agent (`kimi-k2-0905-preview`), and seven web-search-tool-enhanced reasoning models (`gemini-2.5-pro`, `gpt-5-2025-08-07`, `gpt-4o-search-preview-2025-03-11`, `gpt-4.1-2025-04-14`, `claude-opus-4-1-20250805`, `claude-sonnet-4- 20250514`, and `claude-3-7-sonnet-20250219`).

To eliminate the randomness, all models with adjustable temperature were evaluated at temperature = 0.0. For non-DRA models without embedded annotations, reports were merged with annotations during evaluation. `gpt-4o-20-24-11-20` independently scored each rubric to evaluate semantic quality. Topical focus was assessed on pure report text without annotations to avoid interference from annotation titles. Related prompts are provided in Appendix D. Retrieval Trustworthiness was computed from pure annotations, excluding parameters, anchors, and duplicates to ensure fair matching. Manual verification was randomly conducted on approximately 35% of the scores judged by LLMs, yielding a 99.3% agreement with human evaluations.

Regarding parameters, $\alpha = \beta = 0.5$ were set to balance task completion and general quality; $\lambda = 0.7$ and $\mu = 0.3$ were configured to emphasize sensitivity to the omission of FAKs; coefficient was set to $\eta = 0.2$ to control score inflation and prevent over-amplification; $\theta = 0.7$ and $\kappa = 0.3$ were set for the contributions of exact citations and generalized mentions. This configuration ensures scoring stability while enhancing sensitivity to semantic relevance, citation accuracy, and task fidelity.

### 5.2. Leaderboard

Table 1 and Appendix E presents the leaderboard results, with models ranked in descending order of IntegratedScore. In terms of IntegratedScore, `Qwen` ranked first, demonstrating strong performance across all dimensions. `Sonar` followed closely, achieving the highest score in topical focus. Notably, `Kimi-K2`, a Mixture-of-Experts architecture agent with 1T parameters, attained the highest score in the quality, outperforming all DRAs. For topical focus, `Sonar`, `Qwen`, and `o3` formed the leading cluster. `GPT-5` achieved the highest score in citation reliability, indicating strong external support alignment. In terms of resource consumption, `o3` and `o4-mini` averaged 23K and 18K tokens per report respectively, placing them lowest in contribution efficiency.

Overall, DRAs exhibited stable semantic quality and control in structured report generation, while web-search-tool-augmented models showed promise in external support and efficiency. These results validate the DR. BENCH's ability to differentiate model capabilities across multiple dimensions, providing a robust empirical foundation for future optimization of DRAs.

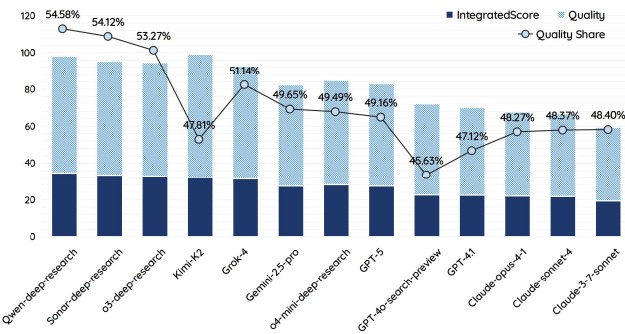

*Figure 3.* The proportion of quality share across models.

Figures 3 and 10 summarize the role of Quality scores and cross-domain performance in the overall IntegratedScore. While Kimi-K2 achieves a strong Quality score, its relative weaknesses in credibility and attention reduce the impact of this advantage, whereas models such as Qwen, Sonar, and o3 exhibit a more balanced interplay between Quality and multiplicative factors, yielding stronger integrated performance. Across domains, all top models perform notably well in domain 03 (Sports & Competitions) and domain 10 (Health & Medicine), while maintaining relatively balanced results elsewhere. In Quality, Kimi-K2 leads, followed by Qwen and Sonar; however, in SemanticDrift and TrustworthyBoost, Kimi-K2 shows clear disadvantages, whereas the others remain comparatively consistent. Detailed analysis is provided in Appendix E.

### 5.3. Supplementary Dimensions

Additionally, four OpenAI models yielded richer response metadata, revealing strategic differences in task execution and tool usage beyond the three core dimensions, and offering valuable supplement into multidimensional evaluation.

*Table 2.* Average inference and retrieval times on OpenAI models.

| Models | ReasonTimes | SearchTimes |
|---|---|---|
| GPT-4.1 | *Unavailable* | 0.3925 |
| GPT-5 | 12.9346 | 10.6729 |
| o3-dr | 55.4333 | 16.1000 |
| o4-mini-dr | 63.9860 | 26.5093 |

As shown in Table 2, GPT-4.1 showed minimal retrieval activity, with an average of only 0.39 times. In contrast, both o4-mini-dr and o3-dr displayed intensive inference and retrieval patterns, suggesting more complex rea-

soning chains and information acquisition strategies. In terms of retrieval efficiency, o3-dr slightly exceeded in total retrieved links and annotations, while o4-mini-dr achieved slightly lower RetrievalIndex, reflecting stronger filtering and citation precision, as shown in Table 3.

*Table 3.* RetrievalIndex of o3-dr and o4-mini-dr.

| Models | num$_{retr}$ | num$_{anno}$ | RIndex ↓ |
|---|---|---|---|
| o4-mini-dr | 14.4583 | 7.7986 | 0.5520 |
| o3-dr | 15.2744 | 8.7561 | 0.5804 |

## 6. Discussions

Two systemic limitations emerged during evaluation that underscore key challenges in the design of DRAs. First, instability in invocation behavior was observed in models such as o3 and o4-mini, which exhibited substantial variance in reasoning time across repeated queries. This suggests a lack of internal constraints governing search frequency and direction, resulting in non-convergent retrieval paths and inconsistent response behavior. Second, semantic decomposition occasionally produced sub-queries in non-English languages with incoherent semantics, despite all tasks being in English. These outputs were misaligned with task intent and unintelligible to human evaluators, thereby impairing retrieval precision and relevance.

These limitations reflect two fundamental trade-offs in DRAs development. The efficiency—quality trade-off highlights the tension between high-quality reasoning and computational cost, with current models often incurring excessive token usage and latency. Addressing this requires adaptive control over search depth and token allocation. Meanwhile, the decomposition—coherence trade-off reveals that while modular query breakdown enhances coverage, it risks semantic fragmentation and intent drift. Future architectures must reconcile decomposition benefits with coherent multi-stage reasoning to ensure consistent task fidelity.

## 7. Conclusions

In this paper, we present the DR. BENCH and the multi-dimensional evaluation framework that systematically assesses the performance and capabilities of DRAs. By leveraging challenging queries across diverse thematic domains and high-quality reference bundles, our framework enables rigorous evaluation of report-style outputs along axes of semantic quality, topical focus, and retrieval trustworthiness. Empirical results show that contemporary DRAs substantially outperform conventional tool-augmented models in complex task scenarios, while also exposing key limitations and trade-offs. These insights elucidate current challenges in DRA design and lay the groundwork for advancing DRAs as efficient, stable, and interpretable intelligent agents.

## Impact Statement

This paper proposes the DR. BENCH for evaluating Deep Research Agents, aiming to advance the field of Machine Learning and Artificial Intelligence. By emphasizing multidimensional evaluation, it promotes transparency, reproducibility, and trustworthiness in AI-generated research. The framework can help mitigate risks of misinformation while supporting responsible adoption of agent-based systems. We believe these contributions align with established trajectories in machine learning, with no specific negative societal consequences requiring further emphasis.

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

## A. Taxonomy of Domains

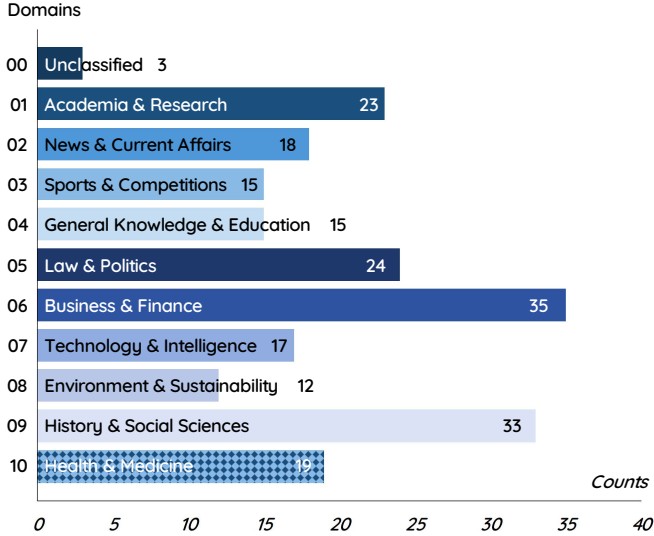

*Figure 4.* Taxonomy of domains.

Figure 4 and Table 4 present the domain taxonomy underlying the benchmark corpus. The classification scheme comprises ten principal domains, delineated according to thematic relevance: `Academia & Research`, `News & Current Affairs`, `Sports & Competitions`, `Commonsense & Education`, `Law & Politics`, `Business & Finance`, `Technology Intelligence`, `Environment & Sustainability`, `History & Social Sciences`, and `Health & Medicine`. Entries that do not align with the predefined domains are assigned to the residual class `Unclassified`, in order to preserve high-quality data while maintaining diversity.

*Table 4.* Taxonomy of domains with simplified descriptions.

| Code | Domain | Description | Count |
|---|---|---|---|
| 00 | Unclassified | Cannot be categorized | 3 |
| 01 | Academia & Research | Academic trends, research methods, etc. | 23 |
| 02 | News & Current Affairs | International news, regional hotspots, etc. | 18 |
| 03 | Sports & Competitions | Olympics, World Cup, athlete data, match reports, etc. | 15 |
| 04 | Commonsense & Education | Common facts, educational resources, etc. | 15 |
| 05 | Law & Politics | Legal texts, policy updates, international relations, etc. | 24 |
| 06 | Business & Finance | Market analysis, investment strategies, etc. | 35 |
| 07 | Technology Intelligence | Artificial intelligence, tech trends, etc. | 17 |
| 08 | Environment & Sustainability | Climate change, environmental policies, ecosystems, etc. | 12 |
| 09 | History & Social Sciences | History, philosophy, sociology, psychology, etc. | 33 |
| 10 | Health & Medicine | Disease research, public health, nutrition knowledge, etc. | 19 |
| ***Summary*** | | | ***214*** |

It is evident that `Business & Finance` and `History & Social Sciences` account for a relatively larger proportion of the benchmark. Nevertheless, the overall distribution remains broadly balanced, exhibiting both thematic richness and representational breadth.

## B. General-Report Rubrics

[ GRRs ]
(1) Does the report include a clear three-part structure (introduction, body, conclusion)? Yes=2, No=0
(2) Does the report clearly state the research question or objective at the beginning? Yes=2, No=0
(3) Does the report provide background and purpose in the introduction? Yes=1, No=0
(4) Does the report develop coherent arguments in the body section? Yes=2, No=0
(5) Does the report summarize key findings in the conclusion? Yes=2, No=0
(6) Does the report offer actionable recommendations or future directions? Yes=2, No=0
(7) Does the report use smooth transitions between paragraphs or sections? Yes=1, No=0
(8) Does the report use headings and subheadings to organize content? Yes=1, No=0
(9) Does the report avoid information dumping and present ideas clearly? Yes=2, No=0
(10) Does the report use precise and clear language to express ideas? Yes=2, No=0
(11) Does the report avoid grammar, spelling, or sentence structure issues? Yes=1, No=0
(12) Does the report demonstrate logical reasoning such as cause-effect or comparison? Yes=2, No=0
(13) Does the report reflect critical thinking or independent judgment? Yes=2, No=0
(14) Does the report conclude with insightful perspectives or calls to action? Yes=1, No=0
(15) Does the report maintain a formal, academic, and objective tone throughout? Yes=1, No=0
(16) Does the report cover all key aspects of the research topic? Yes=2, No=0
(17) Does the report avoid missing important background or variables? Yes=1, No=0
(18) Does the report provide sufficient evidence to support its claims? Yes=2, No=0
(19) Does the report analyze underlying causes or trends in the data? Yes=2, No=0
(20) Does the report incorporate multiple angles or dimensions in its analysis? Yes=1, No=0
(21) Does the report demonstrate both breadth and depth of understanding? Yes=2, No=0
(22) Does the report avoid vague or repetitive statements? Yes=1, No=0
(23) Does the report cite authoritative academic journals or professional sources? Yes=2, No=0
(24) Does the report provide clear citation formatting? Yes=1, No=0
(25) Does the report cite sources that are highly relevant to the topic? Yes=2, No=0
(26) Does the report avoid fabricated, unclear, or misleading references? Yes=2, No=0
(27) Does the report embed citations within the body rather than only at the end? Yes=1, No=0
(28) Does the report distinguish between primary and secondary sources? Yes=1, No=0
(29) Does the report offer a unique perspective or analytical framework? Yes=2, No=0
(30) Does the report critique existing viewpoints thoughtfully? Yes=2, No=0
(31) Does the report propose innovative ideas or future research directions? Yes=2, No=0
(32) Does the report show deep understanding of complex issues? Yes=2, No=0
(33) Does the report avoid simply repeating existing conclusions? Yes=1, No=0
(34) Does the report reflect the author's reasoning and intellectual depth? Yes=2, No=0
(35) Does the report use credible and verifiable data sources? Yes=2, No=0
(36) Does the report interpret and explain data appropriately? Yes=2, No=0
(37) Does the report use charts, tables, or visuals to support analysis? Yes=1, No=0
(38) Does the report avoid misusing statistics or exaggerating findings? Yes=2, No=0
(39) Does the report analyze data with causal or trend-based reasoning? Yes=2, No=0
(40) Does the report acknowledge limitations or biases in the data? Yes=1, No=0
(41) Does the report include source and date information for cited data? Yes=1, No=0
(42) Does the report use proper Markdown heading levels (e.g., #, ##, ###)? Yes=1, No=0
(43) Does the report use ordered or unordered lists to present key points? Yes=1, No=0
(44) Does the report correctly use Markdown elements like code blocks, quotes, or tables? Yes=1, No=0
(45) Does the report avoid Markdown syntax errors or formatting issues? Yes=1, No=0
(46) Does the report maintain clean, readable, and visually consistent layout? Yes=1, No=0
(47) Does the report use consistent terminology and avoid style shifts? Yes=1, No=0
(48) Does the report avoid informal or conversational language? Yes=1, No=0

*Figure 5.* Detailed criteria for General-Report Rubrics.

## C. Examples of Entries

To illustrate the precision and rigor of our benchmark, we present four representative entries: ID 04216, 07001, 08004, and 10236. These entries span distinct domains and query types and serve as concrete examples of structural completeness, rubric coverage, and citation fidelity.

[ UID ]  07001
[ Domain ]  07

[ Query ]  The QUIC transport protocol was published in May 2021. please summarize a report of its standardization path, including a comparison of the different drafts and the final RFC, as well as the substantive technical changes between different RFC versions.

[ QSRs ]  (1) Does the report explicitly enumerate QUIC WG draft milestones aligned to at least eight distinct IETF meetings between 2016 and 2021? Yes=2, No=0
(2) Does the report explicitly and accurately contrast technical changes between draft-17 and draft-23 in handshake flows (0-RTT rules and Initial/Handshake packets)? Yes=2, No=0
(3) Does the report contrast technical changes between draft-17 and draft-23 in packet number spaces? Yes=2, No=0
(4) Does the report correctly describe the scope of RFC 8999 as invariants? Yes=2, No=0
(5) Does the report correctly describe the scope of RFC 9000 as transport? Yes=2, No=0
(6) Does the report correctly describe the scope of RFC 9001 as TLS usage? Yes=2, No=0
(7) Does the report state that RFC 9002 defines loss detection, retransmission timing, and congestion control algorithms? Yes=2, No=0
(8) Does the report correctly state that RFC 9002 specifies NewReno-style congestion control and QUIC-specific loss detection with Probe Timeout (PTO)? Yes=2, No=0
(9) Does the report explicitly state that QUIC was originally proposed by Google? Yes=1, No=0
(10) Does the report explicitly state that draft-34 (2021) was the final draft version? Yes=1, No=0
(11) Does the report explicitly define or expand abbreviations such as QUIC, RFC, and 0-RTT upon first use? Yes=2, No=0
(12) Does the report explicitly mention that in May 2023 the IETF published RFC 9369? Yes=2, No=0
(13) Does the report explicitly mention that draft-13/14 established the "QUIC + TLS 1.3" model? Yes=2, No=0
(14) Does the report explicitly mention that draft-17 introduced independent packet number spaces? Yes=2, No=0
(15) Does the report mention that draft-29 became the widely used baseline for interoperability testing? Yes=2, No=0
(16) Does the report explicitly mention that QUIC standardization represented an evolution in the TCP/IP stack by integrating transport design and encryption protocols? Yes=1, No=0
(17) Does the report explicitly mention that RFC 8999 defined the fields that must remain consistent across versions? Yes=1, No=0

[ TSLs ] https://www.rfc-editor.org/rfc/rfc8999/, https://www.rfc-editor.org/rfc/rfc9000/, https://www.rfc-editor.org/rfc/rfc9001/, https://www.rfc-editor.org/rfc/rfc9002/, https://datatracker.ietf.org/wg/quic/meetings/

[ FAKs ]  0-RTT, TLS 1.3, Long Header, Probe Timeout, NewReno
[ FDKs ]  HTTP/2, DTLS, SCTP, TCP Fast Open, SPDY

*Figure 6.* Example ID 07001 of benchmark entries.

[ UID ] 04216
[ Domain ] 04

[ Query ] I saw a stray cat by the roadside. Please provide me with a report on what preparations are needed to adopt a stray cat, and what I should pay attention to during the first seven days at my home, and the must-dos after adopting a cat from the shelter.

[ QSRs ] (1) Does the report mention what to prepare before taking the cat to the vet (e.g., medical records, stool sample)? Yes=1, No=0
(2) Does the report describe how to initially check at home for fleas or ear mites? Yes=1, No=0
(3) Does the report mention how to assess if the cat has long-term aggression issues? Yes=1, No=0
(4) Does the report explain how to help a cat adapt to interactions with children? Yes=1, No=0
(5) Does the report address how to help a cat adapt to existing pets such as dogs, not just other cats? Yes=1, No=0
(6) Does the report mention what to do if the cat refuses interaction beyond the first 7 days? Yes=1, No=0
(7) Does the report explain how to choose food based on the cat's age? Yes=1, No=0
(8) Does the report explain pros and cons of different bowl materials (stainless steel, ceramic, plastic)? Yes=1, No=0
(9) Does the report explain pros and cons of raw food, wet food, and dry food? Yes=1, No=0
(10) Does the report mention placing the litter box away from noisy or high-traffic areas? Yes=1, No=0
(11) Does the report describe play techniques to build trust with a timid cat? Yes=1, No=0
(12) Does the report mention how to detect separation anxiety in the first week? Yes=1, No=0
(13) Does the report mention the long-term financial costs of cat ownership? Yes=1, No=0
(14) Does the report mention local legal requirements or regulations after adoption? Yes=1, No=0
(15) Does the report include guidelines about litter box setup, clearly stated location hygiene and capacity guidelines (e.g., the n+1 rule, separation from food/water, daily scooping)? Yes=1, No=0
(16) Does the report mention recommending vertical space (cat tree), setting up scratching posts, providing daily interactive play, and safe toys to reduce the cat's stress and support adjustment? Yes=1, No=0
(17) Does the report correctly mention keeping small objects that are usually placed on tables away in case the cat would cause any damage? Yes=1, No=0
(18) Does the report mention the importance of ensuring windows are correctly closed/secured for high-rise housing and explain the reason: to prevent the cat from accidentally falling out of the window? Yes=1, No=0
(19) Does the report mention the preparation of clear, moving water and changing the water regularly? Yes=1, No=0
(20) Does the report correctly mention that internal (deworming schedule) and external (year-round flea/tick prevention) parasite prevention plans are required for the cat? Yes=1, No=0
(21) Does the report mention at least two popular infectious diseases (such as FIV, FIP, Toxoplasmosis) that the cat may catch with credible sources and proper citations? Yes=1, No=0
(22) Does the report mention at least two diseases (Cat Scratch Disease, Toxoplasmosis) and two parasites (Flea, Ringworm, Scabies) that humans can catch from a cat, with detailed cause, transmission, and citation? Yes=1, No=0
(23) Does the report correctly mention microchipping, registering the cat, and equipping the cat with a traceable collar with contacts? Yes=1, No=0
(24) Does the report mention quarantine the cat with a "safe-room" approach and the reason for this (adapt the cat to the environment, etc.) and explicitly mention the 1–2 week cycle? Yes=1, No=0
(25) Does the report mention regularly brushing the cat and providing nail care for the cat in routine? Yes=1, No=0
(26) Does the report mention emphasizing patience, positive reinforcement (no punishment), carrier training/desensitization, and normalizing initial hiding/hissing for behavioral acclimation & socialization? Yes=1, No=0
(27) Does the report mention the basic public health guidance for cat owners with at least three examples (handwashing, litter box hygiene, keep the cat indoors, etc.) with proper reference to the zoonosis awareness of CDC? Yes=1, No=0
(28) Does the report clearly mention the "don't s" with at least two examples, including declawing and unsupervised outdoor free-roaming? Yes=1, No=0
(29) Does the report clearly mention the confirmation or schedule of the cat's spay/neuter, and clearly explain both the medical and behavioral necessity and benefits with examples? Yes=1, No=0
(30) Does the report clearly mention the initial veterinary exam within 3–7 days that reviews the cat's prior records, performs a full physical, fecal/parasite check, and FeLV/FIV testing where appropriate; establishes a vaccine plan (core FVRCP and rabies; FeLV for kittens/at-risk) with correct boosters? Yes=1, No=0

[ TSLs ] https://avmajournals.avma.org/view/journals/javma/260/12/javma.22.03.0109.xml, https://www.cdc.gov/healthy-pets/pets-animals/index.html, https://avmajournals.avma.org/view/journals/javma/239/5/javma.239.5.625.xml, https://www.vet.cornell.edu/departments-centers-and-institutes/cornell-feline-health-center/health-information/feline-health-topics/zoonotic-disease-what-can-i-catch-my-cat, https://www.phoenixvilleanimalhospital.com/health-wellness/feline-infectious-diseases/, https://bestfriends.org/pet-care-resources/new-cat-checklist-welcome-your-new-feline-friend-home, https://www.cdc.gov/healthy-pets/about/cats.html, https://www.pawschicago.org/news-resources/all-about-cats/getting-started-a-guide-for-bringing-home-a-new-cat/introducing-a-new-cat-into-your-household

[ FAKs ] vet, deworming, vaccine, microchip, quarantine
[ FDKs ] feral, breeding, exotic, straydog, wildlife

*Figure 7.* Example ID 04216 of benchmark entries.

[ UID ] 08004
[ Domain ] 08

[ Query ] Summarize the history of international climate change negotiations. Write a report that analyzes the key mechanisms of the Paris Agreement, compares the commitments of developed and developing countries, and evaluates its impact on the transition toward renewable energy and sustainable development.

[ QSRs ] (1) Does the report explicitly state that the United Nations Conference on Environment and Development (UNCED, 1992) was held in Rio de Janeiro, Brazil? Yes=1, No=0
(2) Does the report explicitly state that the 1992 United Nations Framework Convention on Climate Change (UNFCCC) aimed to stabilize greenhouse gas concentrations in the atmosphere to avoid 'dangerous anthropogenic interference' with the climate system? Yes=2, No=0
(3) Does the report explicitly state that the 1992 UNFCCC established the principle of 'common but differentiated responsibilities,' requiring all countries to act but assigning greater responsibility to developed nations? Yes=1, No=0
(4) Does the report explicitly state that the 1997 Kyoto Protocol for the first time set legally binding emission reduction targets for developed countries? Yes=1, No=0
(5) Does the report explicitly state that the United States did not ratify the 1997 Kyoto Protocol, citing the absence of developing-country obligations as a key reason? Yes=1, No=0
(6) Does the report explicitly describe the 'Kyoto mechanisms,' namely International Emissions Trading (IET), the Clean Development Mechanism (CDM), and Joint Implementation (JI)? Yes=2, No=0
(7) Does the report explicitly state the relationship between the UNFCCC (framework principles and objectives) and the Kyoto Protocol (implementation rules)? Yes=1, No=0
(8) Does the report explicitly state that the 2009 United Nations Climate Change Conference (COP15) was held in Copenhagen, Denmark? Yes=1, No=0
(9) Does the report state that COP15 sought a legally binding global agreement but did not produce one? Yes=1, No=0
(10) Does the report explicitly state that the 21st Conference of the Parties (COP21) to the UNFCCC was held from November to December 2015? Yes=1, No=0
(11) Does the report explicitly state that COP21 of the UNFCCC was held in Le Bourget, Paris, France? Yes=2, No=0
(12) Does the report explicitly state that at COP21, 195 Parties to the UNFCCC adopted the Paris Agreement? Yes=1, No=0
(13) Does the report explicitly state that the goal of the Paris Agreement is to limit global warming to well below 2°C, while pursuing efforts to limit it to 1.5°C? Yes=1, No=0
(14) Does the report explicitly state that under the Paris Agreement, Nationally Determined Contributions (NDCs) must be updated every 5 years? Yes=2, No=0
(15) Does the report explicitly state that the 1992 Earth Summit refers to the United Nations Conference on Environment and Development (UNCED)? Yes=1, No=0
(16) Does the report explicitly state that the 2009 Copenhagen Conference refers to the 2009 United Nations Climate Change Conference (COP15)? Yes=1, No=0
(17) Does the report explicitly state that Article 4 of the Paris Agreement establishes the system of Nationally Determined Contributions (NDCs)? Yes=1, No=0
(18) Does the report explicitly state that Article 13 of the Paris Agreement establishes the Enhanced Transparency Framework (ETF)? Yes=1, No=0
(19) Does the report state that Article 14 of the Paris Agreement establishes the Global Stocktake (GST)? Yes=1, No=0
(20) Does the report explicitly state that the operational guidance for Article 6 was finalized at COP26 in Glasgow (completing the Paris Rulebook)? Yes=1, No=0
(21) Does the report explicitly state that under the Paris Agreement, developed countries must adopt economy-wide absolute emission reduction targets? Yes=1, No=0
(22) Does the report explicitly state that under the Paris Agreement, developed countries have obligations to provide finance, technology, and capacity-building support? Yes=1, No=0
(23) Does the report explicitly state that the Paris Agreement's transparency framework and global stocktake together form a 'feedback and accountability' cycle? Yes=1, No=0
(24) Does the report explicitly state that in 2023, around 507–510 GW of new renewable power capacity was added globally, nearly 50% more than in 2022? Yes=1, No=0
(25) Does the report explicitly state that the Paris Agreement has a stronger incentivizing effect in countries with weaker governance capacity? Yes=2, No=0

[ TSLs ] https://unfccc.int/resource/docs/convkp/conveng.pdf, https://www.un.org/en/conferences/environment/rio1992, https://unfccc.int/resource/docs/convkp/kpeng.pdf, https://unfccc.int/kyoto_protocol, https://unfccc.int/process/conferences/pastconferences/copenhagen-climate-change-conference-december-2009, https://unfccc.int/documents/meetings/unfccc_archive, https://unfccc.int/sites/default/files/english_paris_agreement.pdf, https://unfccc.int/process-and-meetings/conferences/past-conferences/paris-climate-change-conference-november-2015/cop-21, https://unfccc.int/process-and-meetings/conferences/glasgow-climate-change-conference-october-november-2021, https://unfccc.int/gla

[ FAKs ] UNCED, CBDR, Transparency Framework, COP21, 1.5°C
[ FDKs ] League of Nations, NATO, carbon tax, industrial revolution, population growth

*Figure 8.* Example ID 08004 of benchmark entries.

[ UID ] 10236
[ Domain ] 10

[ Query ] Please write a report analyzing the implications of global public health crises on international cooperation mechanisms, with a case study of the COVID-19 pandemic.

[ QSRs ] (1) Does the report explicitly provide a quantitative estimate of the economic costs of failed international cooperation during COVID-19? Yes=2, No=0
(2) Does the report clearly compare international cooperation in COVID-19 with past health crises such as Ebola, SARS, or H1N1? Yes=2, No=0
(3) Does the report explicitly analyze the role of non-state actors such as NGOs, religious groups, and grassroots organizations in global cooperation? Yes=2, No=0
(4) Does the report explicitly examine how nationalism and populism shaped public opinion against international cooperation? Yes=2, No=0
(5) Does the report explicitly explore the role of international law in compelling or constraining state behavior during the pandemic? Yes=2, No=0
(6) Does the report specifically provide case studies of successful bilateral cooperation (e.g., vaccine donations, medical aid) and their limitations? Yes=3, No=0
(7) Does the report explicitly evaluate how geopolitical rivalries influenced funding allocations for WHO and COVAX? Yes=2, No=0
(8) Does the report explicitly consider the role of regional rivalries (e.g., India-Pakistan, Gulf states) in shaping cooperation outcomes? Yes=2, No=0
(9) Does the report explicitly consider cooperation failures in equitable vaccine distribution for refugees and stateless populations? Yes=2, No=0
(10) Does the report explicitly analyze cooperation in genomic surveillance beyond the South Africa/Omicron example? Yes=2, No=0
(11) Does the report explicitly assess cooperation on clinical data sharing across borders? Yes=2, No=0
(12) Does the report explicitly analyze the role of intellectual diplomacy (science diplomacy) in easing tensions during COVID-19? Yes=2, No=0
(13) Does the report explicitly analyze the influence of domestic political cycles (elections) on willingness to engage in cooperation? Yes=3, No=0
(14) Does the report explicitly evaluate the role of international education networks (e.g., student exchanges) in sustaining cooperation during COVID-19? Yes=2, No=0

[ TSLs ] https://www.who.int, https://www.gavi.org/covax-facility, https://www.un.org/securitycouncil/, https://www.worldbank.org/en/topic/health

[ FAKs ] World Health Organization, COVAX, vaccine nationalism, international cooperation, Ebola
[ FDKs ] HIV, malaria, non-communicable diseases, climate change and health, poverty alleviation

*Figure 9.* Example ID 10236 of benchmark entries.

## D. Scoring Prompts

As part of the rubric evaluation process, the following prompt was provided to guide scoring consistency and rule adherence.

```
You are a scoring evaluator tasked with assessing the quality of a report generated by a
deep research model.  You will be provided with:
   1.  A report text
   2.  An evaluation rule containing specific scoring criteria and allowed score values
Your task is to:
   - Carefully read the report
   - Evaluate it strictly against the given rule
   - Assign a score based only on the score values defined in the rule
Scoring instructions:
   - Only use the score values explicitly listed in the rule
   - Do not invent intermediate scores or alternative formats
   - Your output must begin with the score in square brackets [], followed by a
   one-sentence reason
Output format example:
   [0] No citations were provided, which violates the requirement.  [2] The report fully
   meets the requirement with clear and relevant details.
Be objective and consistent.  Focus on clarity, completeness, relevance, and adherence to
the rule.
Report text:  {report}
Rule:  {rubric}
```

As part of the keyword relevance evaluation stage, the following prompt was provided to guide consistent scoring.

```
You are a scoring evaluator tasked with assessing the relevance of a specific keyword
within a research report.  You will be provided with:
   1.  A report text
   2.  A keyword to evaluate
Your task is to:
   - Read the report carefully
   - Judge how semantically relevant the keyword is to the report
   - Consider not just frequency, but depth of discussion, thematic importance, and
   contextual integration
Use the following 5-point relevance scale:
   (5) Extremely Relevant:  The keyword is a central theme of the report; It appears
   multiple times and is discussed in depth; The report's main arguments or findings
   revolve around it; (4) Highly Relevant:  The keyword is a major topic; It appears more
   than once and is clearly explained or referenced; contributes directly to the report's
   purpose; (3) Moderately Relevant:  The keyword is mentioned but not emphasized; It
   may appear once or twice; It supports the report contextually but is not a focus; (2)
   Slightly Relevant:  The keyword is briefly mentioned; It has little impact on the
   report's core content; It may be incidental or peripheral; (1) Not Relevant:  The
   keyword does not appear in the report; Or it appears in a way that is unrelated to the
   report's topic.
Output format example:
   [4] The keyword "QUIC" is referenced multiple times in the report, particularly in the
   context of protocol evolution and RFC publication.  While not the sole focus, it is
   clearly a major topic.
Be objective and consistent.  Focus on clarity, completeness, relevance, and adherence to
the rule.
Report text:  {report}
Keyword:  {keyword}
```

# E. Supplementary Experimental Observations

### E.1. Quality Share Proportion

Figure 3 illustrates the contribution of Quality scores to the overall IntegratedScore. As a core metric among the three evaluation dimensions, Quality's share proportion provides a direct indication of each DRA's capacity for textual precision, thematic focus, and control over external referencing. Notably, although `Kimi-K2` achieves a prominent score in the Quality dimension, its relative deficiencies in credibility and attention metrics diminish the extent to which its quality advantage is reflected in the overall score. In contrast, models such as `Qwen`, `Sonar`, and `o3` demonstrate a more balanced allocation between Quality and multiplicative factors, exhibiting great integrative performance while leaving room for further optimization.

### E.2. Evaluation Across Domains

Table 5 presents a fine-grained evaluation of various models across distinct domains. In this table, D denotes the domain, and M refers to the evaluation metrics, including QUA (Quality), SDR (1−SemanticDrift), TBO (TrustworthyBoost), and ITS (IntegratedScore). Each column corresponds to a model represented by an abbreviation: `QWE` (`qwen-deep-research`), `SON` (`sonar-deep-research`), `O3D` (`o3-deep-research-2025-06-26`), `KIM` (`kimi-k2-0905-preview`), `GRO` (`grok-4-0709-search`), `GEM` (`gemini-2.5-pro`), `O4D` (`o4-mini-deep-research-2025-06-26`), `GT5` (`gpt-5-2025-08-07`), `G4O` (`gpt-4o-search-preview-2025-03-11`), `G41` (`gpt-4.1-2025-04-14`), `OPU` (`claude-opus-4-1-20250805`), `SO4` (`claude-sonnet-4-20250514`), and `S37` (`claude-3-7-sonnet-20250219`). This table is designed to highlight model-specific performance variations across multiple dimensions, providing a structured basis for subsequent analysis.

The distribution of ITS values reveals substantial variation in model performance across domains. In particular, `QWE`, `SON`, and `O3D` consistently achieve higher scores across multiple domains. Their ITS values are notably elevated in domain 03, where they reach 41.27, 38.10, and 40.23 respectively, and in domain 10, with corresponding scores of 39.07, 38.86, and 37.68. These results indicate their stable advantages in these specific contexts. In contrast, models such as `S37`, `SO4`, and `OPU` tend to exhibit lower scores across most domains, reflecting a performance floor. This distribution suggests that models vary in their domain sensitivity and adaptability, with certain systems demonstrating enhanced integrative capabilities under domain-specific conditions.

Table 5. Comparative evaluation across domains.

| D | M | QWE | SON | O3D | KIM | GRO | GEM | O4D | GT5 | G4O | G41 | OPU | SO4 | S37 |
|---|---|---|---|---|---|---|---|---|---|---|---|---|---|---|
| 01 | QUA | 0.6273 | 0.6090 | 0.6340 | 0.6705 | 0.6465 | 0.5774 | 0.5690 | 0.5439 | 0.5013 | 0.4690 | 0.4288 | 0.4362 | 0.3800 |
|  | SDR | 0.5184 | 0.5340 | 0.5457 | 0.4664 | 0.5089 | 0.4873 | 0.4826 | 0.4491 | 0.4667 | 0.4715 | 0.4584 | 0.4616 | 0.4765 |
|  | TBO | 1.0209 | 1.0357 | 1.0304 | 1.0373 | 1.0423 | 1.0161 | 1.0351 | 1.0476 | 1.0102 | 1.0021 | 1.0337 | 1.0245 | 1.0209 |
|  | ITS | 33.3293 | 33.8947 | **36.0683** | 32.7436 | 34.4484 | 29.1322 | 28.9792 | 26.8605 | 23.6181 | 22.3786 | 21.3256 | 20.8999 | 18.9583 |
| 02 | QUA | 0.6190 | 0.6331 | 0.6108 | 0.6432 | 0.6077 | 0.5730 | 0.5572 | 0.4939 | 0.5160 | 0.4937 | 0.4457 | 0.4562 | 0.3887 |
|  | SDR | 0.5112 | 0.5282 | 0.4943 | 0.4324 | 0.4760 | 0.4841 | 0.4668 | 0.4276 | 0.4668 | 0.4722 | 0.4508 | 0.4673 | 0.4751 |
|  | TBO | 1.0146 | 1.0131 | 1.0210 | 1.0070 | 1.0222 | 1.0099 | 1.0158 | 1.0126 | 1.0063 | 1.0055 | 1.0175 | 1.0149 | 1.0096 |
|  | ITS | 32.0267 | **34.0071** | 31.8369 | 28.4567 | 30.1611 | 28.1130 | 26.3594 | 22.3089 | 24.4736 | 23.4507 | 20.9942 | 21.8205 | 18.6347 |
| 03 | QUA | 0.6725 | 0.6198 | 0.6462 | 0.7139 | 0.6414 | 0.5513 | 0.5438 | 0.5349 | 0.5211 | 0.4551 | 0.4928 | 0.4579 | 0.4307 |
|  | SDR | 0.5909 | 0.6008 | 0.6053 | 0.5756 | 0.5651 | 0.5545 | 0.5743 | 0.4812 | 0.5316 | 0.5283 | 0.5265 | 0.5719 | 0.5875 |
|  | TBO | 1.0441 | 1.0248 | 1.0195 | 1.0158 | 1.0160 | 1.0062 | 1.0220 | 1.0307 | 1.0065 | 1.0000 | 1.0149 | 1.0151 | 1.0147 |
|  | ITS | 41.2741 | 38.1028 | 40.2341 | **41.8823** | 37.3244 | 31.4358 | 31.6925 | 27.7716 | 28.2281 | 24.3922 | 26.6635 | 26.5932 | 25.6712 |
| 04 | QUA | 0.6123 | 0.6006 | 0.6465 | 0.6368 | 0.5766 | 0.5329 | 0.5488 | 0.5608 | 0.4589 | 0.4530 | 0.4394 | 0.4250 | 0.3690 |
|  | SDR | 0.5155 | 0.4971 | 0.5094 | 0.4789 | 0.4628 | 0.4833 | 0.4807 | 0.4560 | 0.3981 | 0.4401 | 0.4460 | 0.4287 | 0.4259 |
|  | TBO | 1.0108 | 1.0218 | 1.0142 | 1.0132 | 1.0322 | 1.0186 | 1.0163 | 1.0213 | 1.0126 | 1.0065 | 1.0137 | 1.0108 | 1.0219 |
|  | ITS | 31.6576 | 30.1116 | **33.7925** | 31.1660 | 27.8452 | 26.0326 | 26.9029 | 27.1555 | 19.0310 | 20.2277 | 19.8213 | 18.3432 | 16.1287 |
| 05 | QUA | 0.5891 | 0.6012 | 0.5678 | 0.6463 | 0.6066 | 0.5390 | 0.5694 | 0.5517 | 0.4804 | 0.4664 | 0.4435 | 0.4295 | 0.3900 |
|  | SDR | 0.5031 | 0.5142 | 0.4673 | 0.4249 | 0.4728 | 0.4748 | 0.4649 | 0.4461 | 0.4268 | 0.4623 | 0.4676 | 0.4548 | 0.4534 |
|  | TBO | 1.0153 | 1.0270 | 1.0061 | 1.0075 | 1.0288 | 1.0108 | 1.0262 | 1.0301 | 1.0052 | 1.0054 | 1.0187 | 1.0106 | 1.0085 |
|  | ITS | 30.8795 | **32.4488** | 27.2325 | 28.1336 | 30.3342 | 26.6297 | 27.8097 | 25.7082 | 20.3981 | 21.7361 | 21.0149 | 19.8689 | 17.8701 |
| 06 | QUA | 0.6257 | 0.6143 | 0.6039 | 0.6840 | 0.6051 | 0.5214 | 0.5319 | 0.5450 | 0.4816 | 0.4567 | 0.4591 | 0.4520 | 0.4036 |
|  | SDR | 0.5320 | 0.5099 | 0.5205 | 0.4856 | 0.4994 | 0.4701 | 0.4617 | 0.4649 | 0.4382 | 0.4671 | 0.4613 | 0.4739 | 0.4685 |
|  | TBO | 1.0490 | 1.0201 | 1.0106 | 1.0094 | 1.0347 | 1.0079 | 1.0112 | 1.0235 | 1.0035 | 1.0035 | 1.0139 | 1.0137 | 1.0103 |
|  | ITS | **35.9545** | 32.1937 | 32.0019 | 33.8587 | 31.9281 | 24.9650 | 25.6285 | 26.8070 | 21.4294 | 21.5077 | 21.8287 | 21.7307 | 19.1475 |
| 07 | QUA | 0.6221 | 0.6042 | 0.6104 | 0.6358 | 0.5898 | 0.5416 | 0.6181 | 0.5799 | 0.5026 | 0.4566 | 0.4275 | 0.4408 | 0.3859 |
|  | SDR | 0.5118 | 0.5251 | 0.5044 | 0.4735 | 0.4845 | 0.4972 | 0.4958 | 0.4735 | 0.4569 | 0.4729 | 0.4702 | 0.4699 | 0.4624 |
|  | TBO | 1.0577 | 1.0290 | 1.0215 | 1.0262 | 1.0170 | 1.0093 | 1.0266 | 1.0529 | 1.0127 | 1.0014 | 1.0235 | 1.0165 | 1.0131 |
|  | ITS | **33.4141** | 32.6757 | 31.5103 | 30.8186 | 29.5214 | 27.2947 | 31.2982 | 29.2195 | 23.2293 | 21.7192 | 21.0743 | 21.2224 | 18.4858 |
| 08 | QUA | 0.6741 | 0.6338 | 0.5994 | 0.6610 | 0.5945 | 0.5004 | 0.5095 | 0.5806 | 0.4744 | 0.4685 | 0.4504 | 0.4532 | 0.3981 |
|  | SDR | 0.5093 | 0.4833 | 0.4762 | 0.4198 | 0.4145 | 0.4422 | 0.4350 | 0.4252 | 0.4007 | 0.4380 | 0.4085 | 0.4292 | 0.4048 |
|  | TBO | 1.0179 | 1.0175 | 1.0136 | 1.0060 | 1.0219 | 1.0107 | 1.0124 | 1.0158 | 1.0053 | 1.0022 | 1.0153 | 1.0221 | 1.0119 |
|  | ITS | **35.1035** | 30.7007 | 28.8171 | 27.6535 | 25.0956 | 22.2533 | 23.2147 | 25.4842 | 18.9471 | 20.2026 | 18.4119 | 19.7139 | 16.2288 |
| 09 | QUA | 0.6460 | 0.6245 | 0.6149 | 0.6841 | 0.6019 | 0.5558 | 0.5814 | 0.5700 | 0.4832 | 0.5037 | 0.4657 | 0.4650 | 0.4182 |
|  | SDR | 0.5108 | 0.5152 | 0.5182 | 0.4428 | 0.4752 | 0.4771 | 0.4678 | 0.4622 | 0.4336 | 0.4454 | 0.4616 | 0.4704 | 0.4672 |
|  | TBO | 1.0082 | 1.0176 | 1.0091 | 1.0143 | 1.0225 | 1.0128 | 1.0187 | 1.0501 | 1.0053 | 1.0000 | 1.0270 | 1.0230 | 1.0181 |
|  | ITS | **33.5468** | 32.3533 | 31.8783 | 30.6664 | 29.6514 | 26.7788 | 27.3007 | 28.2353 | 21.0317 | 22.0717 | 21.8632 | 22.1229 | 19.6543 |
| 10 | QUA | 0.6666 | 0.6447 | 0.6639 | 0.7129 | 0.6504 | 0.6012 | 0.6218 | 0.6074 | 0.5407 | 0.5155 | 0.4942 | 0.4598 | 0.4148 |
|  | SDR | 0.5514 | 0.5765 | 0.5453 | 0.4967 | 0.5093 | 0.5021 | 0.4982 | 0.4973 | 0.4913 | 0.5101 | 0.5156 | 0.5091 | 0.5149 |
|  | TBO | 1.0507 | 1.0344 | 1.0379 | 1.0148 | 1.0372 | 1.0293 | 1.0190 | 1.0923 | 1.0103 | 1.0019 | 1.0200 | 1.0321 | 1.0185 |
|  | ITS | **39.0697** | 38.8577 | 37.6762 | 36.0924 | 34.9180 | 31.0504 | 31.7808 | 33.4642 | 27.0223 | 26.4282 | 26.0997 | 24.2348 | 22.0003 |
| MIX | QUA | 0.6348 | 0.6184 | 0.6176 | 0.6707 | 0.6130 | 0.5506 | 0.5666 | 0.5560 | 0.4945 | 0.4762 | 0.4559 | 0.4491 | 0.3996 |
|  | SDR | 0.5248 | 0.5271 | 0.5184 | 0.4671 | 0.4890 | 0.4856 | 0.4803 | 0.4593 | 0.4496 | 0.4694 | 0.4674 | 0.4735 | 0.4737 |
|  | TBO | 1.0288 | 1.0238 | 1.0171 | 1.0153 | 1.0283 | 1.0130 | 1.0203 | 1.0383 | 1.0073 | 1.0027 | 1.0202 | 1.0184 | 1.0148 |
|  | ITS | **34.6480** | 33.4668 | 32.9004 | 32.0651 | 31.3490 | 27.3364 | 28.0391 | 27.3312 | 22.5645 | 22.4382 | 22.0047 | 21.7235 | 19.3415 |

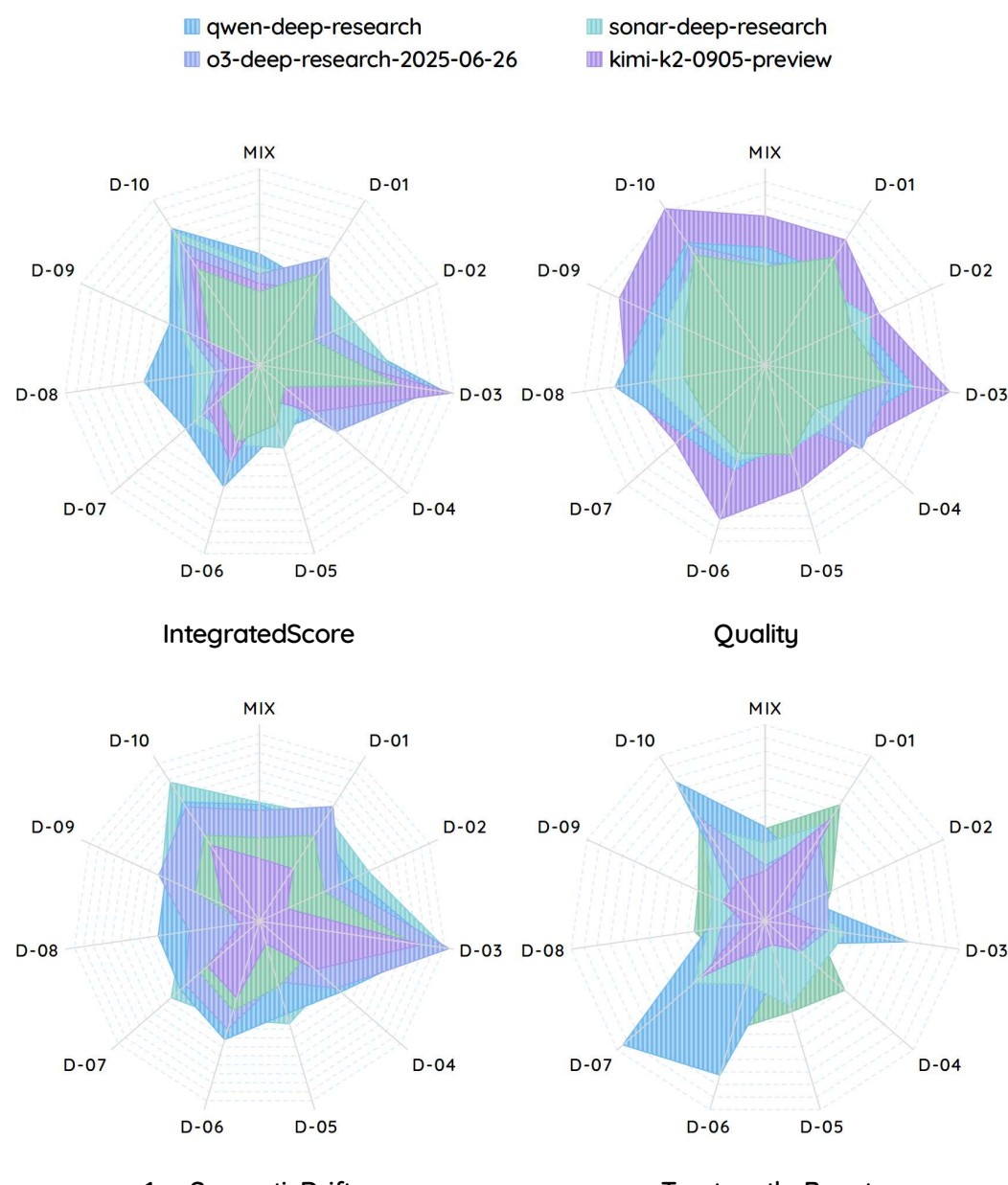

*Figure 10.* Radar charts of top models' performance across domains based on multiple metrics.

Figure 10 presents radar charts illustrating the cross-domain performance of the top five models, with axes rescaled for clarity. In the ITS panel, all models exhibit notably strong performance in domain 03 and domain 10, while maintaining relatively balanced results across other domains. This suggests that the models align most closely with human expectations in the areas of `Sports & Competitions` as well as `Health & Medicine`. In the QUA panel, `KIM` demonstrates a clear advantage, followed by `QWE` and `SON`. Conversely, in the SDR and TBO panels, `KIM` shows a marked disadvantage, whereas the remaining models display comparatively consistent performance.

## F. Comparison with Existing Benchmarks

Table 6 presents a comparison of our benchmark with existing ones across key dimensions. Task Type specifies whether the task is open-ended or closed and whether the target response takes the form of a short string or a long report. Evaluation Dimensions indicate which metrics are applied and which aspects of model behavior are examined. Construction method describes whether the benchmark is primarily built through LLM-based generation or through human annotation. Scale refers to the size of the dataset in terms of the number of instances. Target System specifies whether the benchmark is designed for LLMs equipped with web-search tools or for DRAs. Finally, Evaluation Criteria clarify whether performance is judged by exact matching, LLM-based criteria, or expert rubric assessment.

*Table 6.* Comparison of Benchmarks for Tool-Augmented LLMs

| Benchmarks | Task Type | EvalDim | Constru | Scale | System | EvalCriteria |
|---|---|---|---|---|---|---|
| WebWalker (Wu et al., 2025) | Closed/String | Accuracy | LLMs | 680 | LLMs | Match |
| BrowseComp-Plus (Chen et al., 2025b) | Closed/String | Accuracy | LLMs | 830 | DRAs | Match |
| GAIA (Mialon et al., 2023) | Closed/String | Accuracy | Human | 466 | LLMs | Match |
| BrowseComp (Wei et al., 2025) | Closed/String | Accuracy | Human | 1266 | LLMs | Match |
| WideSearch (Wong et al., 2025) | Closed/String | F1 Score | Human | 200 | LLMs | Match |
| Deep Research Bench (Bosse et al., 2025) | Closed/String | Recall, F1 Score | Human | 89 | LLMs | Match |
| ResearchQA (Du et al., 2025, *Concurrent Work*) | Open/Report | Quality | LLMs | 21K | DRAs | LLMs Criteria |
| ReportBench (Li et al., 2025, *Concurrent Work*) | Open/Report | Quality | LLMs | 678 | DRAs | LLMs Criteria |
| DeepResearch Arena (Wan et al., 2025, *Concurrent Work*) | Open/Report | Quality | LLMs | 10K | DRAs | LLMs Criteria |
| DeepResearch Bench (Du et al., 2025, *Concurrent Work*) | Open/Report | Quality | Human | 100 | DRAs | LLMs Criteria |
| **DR. BENCH (ours)** | **Open/Report** | **Quality, Semantics, Retrieval** | **Human** | **214** | **DRAs** | **Experts Rubrics, Keywords, Links** |

DR. BENCH is the only benchmark that combines quality, semantic adequacy, and retrieval credibility in evaluating report-style outputs. Unlike large-scale auto-generated datasets, it is human-authored with expert rubrics, anchors, and trustworthy links, ensuring transparency and reproducibility. Its 214 carefully curated tasks balance coverage with precision, making it uniquely suited for systematic and fine-grained assessment of Deep Research Agents. Our evaluation dimensions and methodology differ substantially from prior work, with more qualitative and fine-grained details provided in Section 2.2.

## G. Core Dimensions of QSRs

Within the evaluation framework for structured long-text tasks, a critical criterion for assessing the effectiveness and representativeness of QSRs is whether their design demonstrates discriminative capacity in judging task completeness and aligns with human user expectations. QSRs constitute a dimension-based framework tailored to specific queries, intended to measure the completeness of reports. The design rationale of QSRs considers, though is not limited to, the core dimensions outlined in Table 7.

*Table 7.* Core dimensions with descriptions of QSRs

| Dimensions | Descriptions |
| --- | --- |
| Information Coverage | Whether the report fully covers key facts, events, clauses, persons, institutions, or geographic units required by the query, ensuring no essential information is missing. |
| Technical / Mechanism Explanation | Whether the report accurately explains mechanisms, protocols, causal chains, policy tools, or theoretical frameworks involved in the query, reflecting mastery of core logic. |
| Structural Expression | Whether the report generates structured forms such as tables, timelines, maps, matrices, or lists to support organization and reproducibility of the task. |
| Semantic Precision | Whether the report defines key terms, distinguishes concepts, annotates units and time, and avoids ambiguity, ensuring clarity and consistency of expression. |
| Source Verification | Whether the report cites authoritative texts, identifiers, database versions, retrieval times, links, or snapshots to support traceability and credibility of claims. |
| Evidence Organization | Whether the report employs multi-source cross-validation, independent source pairing, counter-factual baselines, placebo tests, or robustness checks to enhance reliability. |
| Heterogeneity and Comparative Analysis | Whether the report identifies and analyzes differences across regions, groups, time periods, or institutions, supporting comparative and explanatory power. |
| Methodological Transparency | Whether the report clearly explains analytical methods, identification strategies, variable definitions, and data processing steps, ensuring consistency and reproducibility. |
| Social Impact and Distribution Effects | Whether the report analyzes distributional impacts of policies or events on different groups, communities, industries, or ecosystems, reflecting social dimensions. |
| Historical Evolution and Temporal Logic | Whether the report establishes event chains, policy evolution paths, time windows, or lag mechanisms, supporting historical and dynamic analysis. |
| Interdisciplinary Integration | Whether the report integrates perspectives from multiple disciplines (e.g., law, economics, technology, society, philosophy) to support complexity and explanatory depth. |

