# OpenReview forum: "Dr. Bench: A Multidimensional Evaluation for Deep Research Agents, from Answers to Reports"
_ICML.cc/2026/Conference — Submitted to ICML 2026_

### Official Review · Reviewer_w36J · 2026-03-12

**Soundness:** 3
**Presentation:** 3
**Significance:** 3
**Originality:** 3
**Overall Recommendation:** 4
**Confidence:** 4

**Summary:**

This paper presents Dr.Bench, an evaluation framework to assess Deep Research Agents (DRAs) from multiple perspectives on their generated research reports. Dr.Bench is composed of a high quality dataset with challenging tasks that spans multiple domains, and it incorporates an evaluation framework that focuses on various aspects of DRA performance, including reasoning quality, trustworthiness, and overall report quality. Experiments on various DRAs presents a preliminary leaderboard-style ranking of different models, showing a consistent outperformance of DRAs against conventional tool-augmented models.

**Compliance With Llm Reviewing Policy:**

Affirmed.

**Final Justification:**

This work presents Dr.Bench, a novel evaluation framework to assess Deep Research Agents (DRAs) from multiple perspectives on their generated research reports. It's strengths include a multidimensional evaluation, a high quality dataset, and extensive evaluation among a variety of models. During the rebuttal phase, the addition of failure modes and the analysis on behavioral tradeoffs between different model architectures further enhances the quality of this work, which fully resolved my raised concerns during the rebuttal phase. Based on all these endavors from the author, I deems this as a qualified work to be accepted to this conference.

**Key Questions For Authors:**

Q1: I would really appreciate it if the authors could provide some analysis on typical scenarios when DRA models fail to generate a high quality report with some potential justifications: Are these failures mainly caused by errors such as citation hallucinations or logical fragmentation during multi-stage reasoning? Such analysis would provide deeper diagnostic insights and further enhance the paper’s contributions.

Q2: The paper would benefit from analyzing how different architectural paradigms, like Kimi-K2’s MoE versus the reasoning chains of o3, lead to specific behavioral trade-offs in retrieval and quality.

Q3: Investigating the specific types of redundant reasoning or non-convergent retrieval paths that lead to excessive token consumption in models like o3 and o4-mini would strengthen the efficiency analysis.

**Limitations:**

Yes

**Strengths And Weaknesses:**

Strengths:
- Multidimensional Evaluation System: Unlike previous benchmarks that focus on short-text or single-dimension outputs, Dr.Bench evaluates across three core dimensions: Semantic Quality, Topical Focus, and Retrieval Trustworthiness.
- High-Quality, Expert-Curated Data: The benchmark covers a decent amount of challenging tasks with diverse domains. Each task is accompanied by a manually constructed "reference bundle" containing Query-Specific Rubrics (QSRs), General-Report Rubrics (GRRs), and Trustworthy-Source Links (TSLs).
- Novel Scoring Mechanisms: The framework introduces SemanticDrift to measure sematic deviation and TrustworthyBoost to reward the use of authoritative, verifiable sources.
- Extensive Experimental Validation: Large-scale experiments involving 13 mainstream models (including DRAs and tool-augmented LLMs) demonstrate the benchmark's ability to differentiate model capabilities with high human agreement (99.3%).

Weaknesses:
- Insufficient Analysis of Error Patterns: While the study identifies general limitations like invocation instability and incoherent sub-queries, it lacks a systematic and granular analysis of specific model failure modes. There is limited quantitative breakdown of why models fail, such as specific categories for hallucinations, logic gaps, or retrieval loops.
- Reliance on Time-Sensitive Links: Although the authors use hostname matching and planned snapshots to mitigate link expiration, the dynamic nature of the web poses a persistent challenge to the long-term stability and reproducibility of retrieval-based metrics.
- High Evaluation Costs: Assessing Deep Research Agents (DRAs) is expensive; models like o3-dr consume an average of 23K tokens per report. The complexity of evaluating multiple rubrics (at least 8 QSRs per query) further adds to the computational and financial burden, which hinders effective replication.
- Systemic Instability of DRAs: Experiments revealed that top models like o3 and o4-mini exhibit high variance in reasoning time and non-convergent retrieval paths. This behavioral inconsistency can make it difficult to achieve stable, repeatable evaluation results.

---

> ### Author Rebuttal · Authors · 2026-03-29
>
> **Dear Reviewer w36J,**
>
> May our response reach you at a favorable moment. We deeply appreciate your feedback, and are willing to clarify the remaining concerns.
>
> ---
>
> **(1) Error analysis of DRA failure modes.**
>
> As most DRAs are closed-source, the complete reasoning path is inaccessible. We strive to infer potential causes through retrospected logs.
>
> Quantitatively, we devised a simple existence test for the following potential fault modes (normalized to [0,1]).
>
> |Modes|o3-deep-research|o4-mini-deep-research|Kimi-K2|
> |---|---|---|---|
> |Citation Mismatch (CM)|0.62|0.49|0.65|
> |Logical Fragmentation (LF)|0.02|0.02|0.00|
> |Retrieval Loop (RL)|0.32|0.35|N/A|
> |Reasoning Jump (RJ)|0.14|0.24|0.17|
> |Language Drift (LD)|0.04|0.07|0.01|
> |Semantic Inconsistency (SI)|0.07|0.10|0.05|
> |Hallucinated Facts (HF)|0.72|0.57|0.70|
> |Redundancy (RD)|0.38|0.35|0.08|
>
> - CM and HF remain elevated, indicating unreliable sourcing and severe hallucinations;
> - RL reveals excessive subquery cycling, resulting in stalled retrieval;
> - RJ signals discontinuities in inference, potentially undermining consistency;
> - OpenAI models show slightly higher LD, with subqueries prone to inconsistent outputs such as Spanish, Serbian, or corrupted text, exposing vulnerabilities in linguistic coherence;
> - RD is comparatively pronounced, leading to resource waste.
>
> Qualitatively, we investigated the major error items of o3‑deep‑research, with cases including:
> - Reports citing irrelevant Wikipedia links, with some Google Scholar references possibly fabricated or unrelated;
> - Multiple subqueries repeatedly searching “2025 Tsinghua graduate ceremony”, producing large‑scale loops;
> - Reports filled with fabricated facts, e.g., “Japan allocated 0.71% of GDP to AI R&D during 2021–2023”, with no source support.
>
> Further analyses will be added to the appendix.
>
> **(2) Difference between the behavioral trade‑offs of o3 and Kimi.**
>
> Since o3 is closed‑source, its detailed reasoning is inaccessible. We therefore attempt to characterize the thought process of Kimi and o3 based on their final reports. Using LLMs, we extracted thought process from the full logs and systematically compared the responses in terms of structure, logic, framework, presentation, content quality, detail depth, and retrieval capability.
> - With 98.0% record support: o3 proves more comprehensive in retrieval scope and citation volume, offering abundant external sources, hyperlinks, and authoritative data with broad coverage and detailed information;
> - With 81.6%: o3 tends toward a narrative and academic style, unfolding logic broadly across background, detail, and analysis;
> - With 71.6%: o3 outputs are lengthy, dense, and costly to read;
> - With 59.7%: Kimi favors concise, structured retrieval, with fewer but more focused citations, emphasizing rapid access to key information;
> - With 97.0%: Kimi’s presentation are more structured, with clear logic, often using bullet points, tables, or timelines to highlight essentials;
> - With 80.1%: Kimi emphasizes practicality and operability, offering concrete action plans and schedules.
>
> We deduce that o3 suits scenarios needing broad background and deep analysis, while Kimi fits goal‑focused contexts with concise, structured, and practical output.
>
> **(3) High evaluation costs of DRA assessment, and causes of excessive token consumption.**
>
> Frankly, high cost is common for DRAs. Since current DRAs deliberately sacrifice tokens for multi‑round analysis and retrieval to produce comprehensive reports, this design distinguishes them from traditional LLMs. In o3’s experiment, the average token count of input and retrieval expansions was 50K; internal reasoning consumed 19K tokens; and the final output was 25K. Notably, intermediate consumption accounts for about 73% of the total. As earlier analysis suggests, potential over‑consumption may stem from factors such as retrieval loop and redundancy.
>
> **(4) Systemic instability of DRAs and reliance on time-sensitive TSLs.**
>
> Regrettably, the instability of DRAs is inherent, especially in web search, where online resources change rapidly and reasoning paths are highly sensitive to retrieved content. Unlike traditional LLMs, evaluation of DRAs concerns whether they can capture dynamic environments to generate detailed reports aligned with human expectations, which is precisely the purpose of Dr. Bench and its multidimensional metrics.
>
> Our dedication to TSLs longevity is rigorous. Only a very small fraction of cases, to preserve diversity and practicality, rely on potentially migratory references. For these, we conduct regular validity tests and perform equivalent replacements with update logs, while providing one‑click update scripts so researchers can reproduce and refresh timestamped versions. This represents perhaps the best trade‑off between effective evaluation and reproducibility.
>
> ---
> We once again express our heartfelt gratitude for your affirmation.
>
> Affectionate and deep homage,
>
> **Authors of Submission 9325**

---

> > ### Author Rebuttal · Reviewer_w36J · 2026-03-31
> >
> > The rebuttal address my concerns and I am willing to raise my score.

---

> > > ### Author Response · Authors · 2026-04-02
> > >
> > > **Dear Reviewer w36J,**
> > >
> > > Thank you for your recognition. We are honored to benefit from your guidance, and your feedback enables us to conduct more precise and in‑depth analyses. We sincerely wish you continued success and fulfillment in your research journey and future endeavors.
> > >
> > > All affectionate good wishes, and most delighted thanks, once again,
> > >
> > > **Authors of Submission 9325**

---

### Official Review · Reviewer_ZwdP · 2026-03-13

**Soundness:** 3
**Presentation:** 3
**Significance:** 4
**Originality:** 3
**Overall Recommendation:** 5
**Confidence:** 3

**Summary:**

This work presents a benchmark dataset for deep research agents. The benchmark consists of 214 tasks curated by experts. It also includes multiple metrics, including semantic quality, topical focus, and retrieval trustworthiness.

The author tested both deep research agents and tool-augmented models on the benchmark set and found that the former outperforms the latter.

**Compliance With Llm Reviewing Policy:**

Affirmed.

**Final Justification:**

Thanks for the response. I will keep my score.

**Key Questions For Authors:**

Are the DRA performances different in some dimensions from their performance on prior benchmark tasks? Or do the performances echo those in the prior benchmark?

**Limitations:**

yes

**Strengths And Weaknesses:**

Strength

The study is designed, situating in the literature; the benchmark is designed to fill in the gaps current benchmark datasets have, such as key agent behaviors are not measured, and discrete, short answers are prevalent, etc.

Another strength is that human experts in multiple domains are deeply involved in creating tasks and data.

Also, the benchmark is sensitive enough to differentiate deep research agents and tool-augmented models, as it has a suite of metrics.

Weakness

I wish the author had compared its benchmark with a couple of prior ones.

---

> ### Author Rebuttal · Authors · 2026-03-29
>
> **Dear Reviewer ZwdP,**
>
> May our response reach you at a favorable moment. We deeply appreciate your time and effort, and are more than willing to address the following confusions.
>
> ---
>
> **(1) Comparison of Dr. Bench with prior ones.**
>
> In fact, prior work has been briefly outlined in Section 2.2 and Appendix F. Here we are pleased to provide a more detailed account of our innovations, framed through two task types, Close/String and Open/Report.
>
> |Bench|TaskType|EvalDimension|Construction|Scale|EvalCriteria|
> |---|---|---|---|---|---|
> |WebWalker|C/S|Accuracy|LLMs|680|Match|
> |GAIA|C/S|Accuracy|Human|466|Match|
> |BrowseComp|C/S|Accuracy|Human|1266|Match|
> |WideSearch|C/S|F1 Score|Human|200|Match|
> |Deep Research Bench|C/S|Recall, F1|Human|89|Match|
> |ResearchQA|O/R|Quality|LLMs|21K|LLMs Criteria|
> |ReportBench|O/R|Quality|LLMs|678|LLMs Criteria|
> |DeepResearch Arena|O/R|Quality|LLMs|10K|LLMs Criteria|
> |DeepResearch Bench|O/R|Quality|Human|100|LLMs Criteria|
> |Dr. Bench (ours)|O/R|Quality, Semantics, Retrieval|Human|214|Experts Rubrics, Keywords, Links|
>
> - Close/String:
>
>   Traditional Close/String benchmarks such as WebWalker, GAIA, BrowseComp, and WideSearch are primarily designed around closed short‑text queries, requiring models to produce verifiable concise outputs. Deep Research Bench targets multi‑step web research, but, still limited to discrete short‑text evaluation, failing to assess structured reports or long‑form analysis, which thus hampers its ability to capture DRAs’ true performance.
>
>   Furthermore, these benchmarks rely heavily on surface matching or similarity measures, which are insufficient for open‑ended tasks. In long‑form generation involving multi‑hop reasoning and cross‑source synthesis, semantic coverage and logical consistency are poorly represented, making it difficult to evaluate the depth and structural quality of model outputs effectively.
>
> - Open/Report:
>
>   To address these limitations, concurrent research has begun exploring Open/Report evaluations. DeepResearch Bench relies heavily on static reference reports, with automated scoring criteria that are vague, overly generalized, and excessively emphasize surface‑level dimensions rather than task completion, failing to reflect human preferences and expectations for report quality and structure. In terms of retrieval, it focuses on consistency between cited passages and reference links, rather than the more critical aspects of credibility and authority.
>
>   ResearchQA, DeepResearch Arena, and ReportBench depend on LLM‑generated rubrics, which struggle to capture structural depth and semantic fidelity. As a result, their evaluations are unstable and lack interpretability, undermining support for high‑precision quality judgments. Moreover, their large scale raises evaluation costs and computational overhead, limiting practical value for real‑world DRAs.
>
> Overall, prior and concurrent benchmarks fail to comprehensively, rigorously, and multidimensionally evaluate DRAs’ long‑form report generation in ways aligned with human expectations, as they lack precise scoring standards and reliable reference systems, making it difficult to systematically characterize DRAs’ integrated capabilities. In contrast, Dr. Bench is an expert‑authored, moderately scaled benchmark tailored to long‑form, report‑style generation, balancing quality, normativity, and efficiency, with a clear, richly dimensioned, systematic, and reproducible evaluation framework designed to fit human expectations of response content rather than relying on mechanized scoring.
>
> **(2) DRA performance on prior benchmarks.**
>
> According to blogs and technical reports released by OpenAI and Qwen, DRAs often excel on prior benchmarks compared to traditional LLMs, owing to their superior tool utilization and integration. This implies that benchmarks originally designed for LLMs have become overly simplistic for DRAs, no longer sufficient to assess agent outputs transitioning from answers to reports. Moreover, relative to exceptional tool-augmented LRMs, conventional benchmarks such as GAIA and WebWalker fail to capture meaningful distinctions. These constitute the rationale and motivation for Dr. Bench’s design, while also highlighting its timeliness.
>
> Dr. Bench is the first high‑difficulty, multi‑domain benchmark meticulously curated by human experts, comprising 214 complex report‑style queries that paired with repeatedly validated reference bundles, enabling a systematic multidimensional evaluation framework tailored to report‑oriented responses. This framework spans critical stages of Deep Research, while introducing joint modeling metrics for semantic quality, topical focus, and retrieval confidence. In doing so, it effectively overcomes the scoring limitations of prior benchmarks and fills existing gaps in assessment dimensions and response formats.
>
> ---
> We once again express our heartfelt gratitude for your affirmation.
>
> Affectionate and deep homage,
>
> **Authors of Submission 9325**

---

> > ### Author Rebuttal · Reviewer_ZwdP · 2026-04-05
> >
> > Thanks for the response. I will keep my score.

---

> > > ### Author Response · Authors · 2026-04-07
> > >
> > > **Dear Reviewer ZwdP,**
> > >
> > > We deeply value your encouragement. Your feedback affords us the chance to further accentuate our contributions and the distinctiveness of our work. We send our warmest wishes for profound satisfaction and enduring joy to accompany you all the way.
> > >
> > > All affectionate good wishes, and most delighted thanks, once again,
> > >
> > > **Authors of Submission 9325**

---

### Official Review · Reviewer_Ai6X · 2026-03-15

**Soundness:** 3
**Presentation:** 3
**Significance:** 3
**Originality:** 3
**Overall Recommendation:** 4
**Confidence:** 2

**Summary:**

The paper introduces Geometry-Guided Relational Multi-Task Learning (GRMT), a framework designed for multi-task learning (MTL) scenarios where supervision is fully non-overlapping. The authors address a critical challenge in MTL. Without shared annotations, task-specific representations often suffer from affine distortions and coordinate ambiguity, which hinders effective knowledge transfer. To solve this, GRMT adopts a geometric perspective. Instead of matching raw feature coordinates, it models each task as a latent geometric structure composed of task-specific anchors and principal directions. The framework uses a shared relation mapper to align the intrinsic relational coordinates  of samples across tasks. This allows the model to synchronize the geometric signatures of samples in a coordinate-invariant space. Evaluated on five genomic sequence prediction tasks, GRMT consistently outperforms single-task and existing partial-label multi-task baselines.

**Compliance With Llm Reviewing Policy:**

Affirmed.

**Final Justification:**

All of my concerns are resolved. I keep my original score, weak accept.

**Key Questions For Authors:**

- Given that GPT-4o is used to independently score the rubrics and assess keyword relevance, did you observe any systematic self-preference bias when evaluating other OpenAI models (like o3-deep-research or o4-mini) compared to models like Qwen or Claude? How much might the IntegratedScore shift if a different LLM judger was employed?

- While the methodology intelligently mitigates link rot by favoring authoritative domains and using hostname matching, how do you plan to scale and maintain the 214 tasks over time as real-world facts and live URLs inevitably evolve?

**Limitations:**

yes

**Strengths And Weaknesses:**

## Strengths
- The benchmark's construction is highly rigorous, utilizing a multi-stage pipeline of expert design, LLM auditing, and three rounds of manual cross-review to ensure high task difficulty and structural integrity.

- This work is highly timely. As the field rapidly shifts toward test-time scaling and more autonomous Large Reasoning Models, standard short-form benchmarks fail to capture the nuances of agentic workflows like task decomposition and cross-source aggregation. DR. BENCH successfully addresses this critical evaluation gap.

- The paper is exceptionally well-structured and clear. The inclusion of the domain taxonomy, detailed rubric prompt examples, and cross-domain radar charts effectively communicates the framework's utility and the models' comparative strengths.

## Weaknesses
- The evaluation relies on GPT-4o as the judge for both the QSRs/GRRs and keyword relevance. While the authors report a 99.3% human agreement rate on a 35% validation subset, the potential for self-preference bias (e.g., favoring the stylistic outputs of OpenAI models) in open-ended generation is not thoroughly analyzed.

---

> ### Author Rebuttal · Authors · 2026-03-29
>
> **Dear Reviewer Ai6X,**
>
> May our response reach you at a favorable moment. We deeply appreciate your feedback, and are willing to clarify the remaining concerns.
>
> ---
>
> **(1) Possible self-preference bias due to reliance on judge model.**
>
> According to Liu’s research [1], GPT’s scoring aligns most closely with human judgment. Since one of our aims is to evaluate if DRA reports are consistent with human expectations, the GPT series was adopted as the judger in our experiments. Log analysis revealed no significant self-preference bias in OpenAI models within our benchmark. To substantiate this, we re-scored outputs of representative o3-deep-research and Qwen-deep-research using Gemini-3-Flash-Preview and Claude-Haiku-4-5-20251001 as judgers, applying full-batch non-random sampling under identical experimental settings. The table below presents the average scores for metrics requiring LLM-as-Judger.
>
> |Judger|o3_GRR_score|o3_QSR_score|o3_FAK_rele|o3_FDK_rele|Qwen_GRR_score|Qwen_QSR_score|Qwen_FAK_rele|Qwen_FDK_rele|
> |---|---|---|---|---|---|---|---|---|
> |GPT|61.9143|11.6095|3.7305|1.5314|66.9439|10.5794|3.6720|1.5981|
> |Gemini|62.5095|10.5571|3.7514|1.5124|65.7103|10.3551|3.8112|1.6224|
> |Claude|60.7571|10.8952|3.8410|1.5590|65.8692|10.4766|3.8421|1.5981|
>
> Respectively, we calculate the sum scores for GRR and QSR under the unnormalized perspective, where the broader range provides clearer interpretive contrast. In comparison with GPT, Gemini and Claude exhibit deviations of about 1, maintaining high consistency. The discrepancies in mean FAK relevance and mean FDK relevance are negligible, both at the 0.01 scale.
>
> To more clearly delineate the impact of this bias, we further compute the corresponding Quality and Semantic scores under different Judgers.
>
> |Judger|o3_Quality|o3_Quality_diff|o3_Semantic|o3_Semantic_diff|Qwen_Quality|Qwen_Quality_diff|Qwen_Semantic|Qwen_Semantic_diff|
> |---|---|---|---|---|---|---|---|---|
> |GPT|0.6176|-|0.5187|-|0.6348|-|0.5248|-|
> |Gemini|0.6041|-0.0135|0.5208|0.0021|0.6227|-0.0121|0.5324|0.0076|
> |Claude|0.5977|-0.0199|0.5256|0.0069|0.6258|-0.0090|0.5331|0.0083|
>
> It can be observed that the absolute error of Quality scores remains at the 0.01 scale, while that of Semantic scores stays at the 0.001 scale. Finally, through controlled verification across different Judger configurations (all new Judger, only Quality, or only Semantic), the integrated score of Qwen consistently surpasses that of o3. We conclude that Gemini and Claude exhibit high consistency with GPT’s scoring on o3, indicating no evident homology bias in GPT.
>
> We posit that this outcome benefits from the original design principles of Dr. Bench: the scoring criteria are highly explicit and rigorously distinguished, with even “partial” conditions clearly defined. For example, one QSR of entry 02118 on the U.S. election is: “Does the report quantify polling errors in key swing states (PA, MI, WI) using raw data ...? Yes=2, Partial=1 (quantifies for some states but not all three), No=0”. This structure effectively mitigates the influence of Judger bias on the final scores, thereby safeguarding the rigor and consistency of Dr. Bench.
>
> > [1] Liu, X.,  Zhu, Y., Lan, Y., Yang C., and Qiao Y. Query-relevant images jailbreak large multi-modal models. arXiv preprint arXiv:2311.17600, 2023.
>
> **(2) Maintenance and expansion of TSLs over time.**
>
> In fact, our dedication to the long-term validity of TSLs is stricter than might be expected. For the majority of data, the initial screening criterion during design was whether there exists absolutely authoritative and unique reference URLs, such as a government or organizational official site, and the queries were tailored to highlight the authority of these URLs at review stage. Only a small portion of cases are introduced to preserve diversity and better reflect real-world conditions, using potentially migratory references. For these, we have established a long-term maintenance plan: TSLs will be tested regularly for validity, including accessibility and substitutability. Once a TSL becomes invalid, for example due to an official site migration, we will perform the most equivalent replacement and update. Replacement logs will be maintained so researchers can reproduce and update versions at specific timestamps.
>
> For users, we will release the code, which will automatically save model responses into log files. We are considering providing one‑click update scripts for potential TSL changes, so that by simply supplying the log file address one can conveniently retrieve the TSL scores at a specified timestamp. Given that the search capability of DRAs is not static and their retrieval process naturally evolves with changes in internet resources, we believe this maintenance strategy represents the best trade-off between effective evaluation and reproducibility.
>
> ---
> We once again express our heartfelt gratitude for your affirmation.
>
> Affectionate and deep homage,
>
> **Authors of Submission 9325**

---

> > ### Author Rebuttal · Reviewer_Ai6X · 2026-04-03
> >
> > All concerns are resolved. I keep my score.

---

> > > ### Author Response · Authors · 2026-04-03
> > >
> > > **Dear Reviewer Ai6X,**
> > >
> > > We sincerely appreciate your commendation. Your suggestions provided an opportunity to further clarify and highlight the key aspects of the rigorous design of our evaluation. We sincerely wish good fortune and achievement accompanying you as you forge ahead.
> > >
> > > By the way, there might be a mistaken paste in the summary section of the review, while the other parts are correct. Please kindly verify.
> > >
> > > All affectionate good wishes, and most delighted thanks, once again,
> > >
> > > **Authors of Submission 9325**

---

### Decision · Program_Chairs · 2026-04-30

**Decision:**

Reject

**Comment:**

The paper introduces Dr. Bench, a benchmark of 214 expert-curated tasks and a multidimensional evaluation framework for long-form reports generated by Deep Research Agents.

## Strengths
- Rigorous, expert-driven benchmark construction with multi-stage auditing and cross-review.
- Timely contribution that fills a clear gap left by short-form, single-dimension benchmarks.
- Multidimensional scoring paired with carefully constructed reference bundles (rubrics, keywords, trustworthy links) enables meaningful differentiation across models.

## Weaknesses
- Reliance on an LLM judge raises concerns about self-preference bias in open-ended scoring.
- Dependence on live web links threatens long-term stability and reproducibility of retrieval-based metrics.
- Inherent variance in DRA execution (non-convergent retrieval, high token usage) complicates repeatable evaluation.
- Limited direct empirical comparison to prior benchmarks in the original submission.

The rebuttal to some extend addressed the judge-bias concern with cross-judge experiments and added useful failure-mode analysis and benchmark comparisons. However, there are still concerns regarding reproducibility and LLM-based scoring. The authors are encouraged to fully address the concerns in the next version.